

# Radiative effects of ozone waves on the Northern Hemisphere polar vortex and its modulation by the QBO

Vered Silverman[1], Nili Harnik[1], Katja Matthes[2], Sandro W. Lubis[3], and Sebastian Wahl[2]

[1]Department of Geophysical, Atmospheric and Planetary Sciences, Tel Aviv University, Tel Aviv, Israel
[2]GEOMAR Helmholtz Centre for Ocean Research Kiel, Kiel, Germany
[3]Department of the Geophysical Sciences, The University of Chicago, USA

*Correspondence to:* Vered Silverman (veredsil@post.tau.ac.il)

**Abstract.**

The radiative effects induced by including interactive ozone, in particular, the zonally asymmetric part of the ozone field, have been shown to significantly change the temperature of the NH winter polar cap, and correspondingly the strength of the polar vortex. However, there is still a debate on whether this effect is important enough for climate simulations to justify the numerical cost of including chemistry calculations in long climate integrations. In this paper we aim to understand the physical processes by which the radiative effects of including interactive ozone, and in particular the radiative effects of zonally asymmetric ozone anomalies (ozone waves), amplify to significantly influence the winter polar vortex. Using the NCAR Whole Atmosphere Community Climate Model in the natural configuration, in which ozone depleting substances and green house gases are fixed at 1960's levels, we find a significant effect on the winter polar vortex only when examining the QBO phases separately. Specifically, the seasonal evolution of the midlatitude signal of the QBO - the Holton–Tan effect - is delayed by one to two months when radiative ozone wave effects are removed. Since the ozone waves affect the vortex in an opposite manner during the different QBO phases, when we examine the full time series, besides an early fall direct radiative effect, we find no statistically significant winter effect. We start by quantifying the direct radiative effect of ozone waves on temperature waves, and consequently on the zonal mean zonal wind, and show that this effect is most significant during early fall. We then show how the direct radiative effect amplifies by modifying the evolution of individual upward planetary wave pulses and their induced mean flow deceleration during early winter when stratospheric westerlies just form and waves start propagating up to the stratosphere. The resulting mean-flow differences accumulate during fall and early winter, after which they get amplified through wave-mean flow feedbacks. We find that the evolution of these early-winter upward planetary wave pulses and their induced stratospheric zonal mean flow deceleration are qualitatively different between QBO phases, providing a new mechanistic view of the extratropical QBO signal (the Holton-Tan effect). We further show how these differences result in an opposite effect of the radiative ozone wave perturbations on the mean flow deceleration for east and west QBO phases.

## 1  Introduction

Chemistry climate models (CCMs), which calculate ozone interactively and therefore include asymmetric ozone effects, exist since the early 2000s (CCMVal, 2010). Due to their large numerical cost, CCMs have mostly been used for stratospheric pro-





cesses only and only in recent years they have been coupled to an interactive ocean for the purpose of performing multi decadal climate simulations. There is still an ongoing debate whether interactive atmospheric chemistry, which is computationally very expensive to run for long-term climate integrations, is required in order to generate an appropriate climate signal. The majority of the fifth Coupled Model Inter-comparison Project (CMIP5) models (Taylor et al., 2012) does not use interactive atmospheric

chemistry, instead they prescribe a zonal mean monthly mean ozone field, thus neglecting the effects of zonal asymmetries in the ozone field (ozone waves). In the upcoming CMIP6 exercise (Eyring et al., 2016) more climate models will perform simulations which will include atmospheric chemistry, however, a majority will still use prescribed ozone fields. In order to compare and evaluate the performance of models using either interactive chemistry (including ozone waves) or prescribed zonal mean ozone (neglecting ozone waves), it is crucial to understand the impact of ozone waves on stratospheric dynamics.

The effects of ozone waves can be formulated as an effective change in the temperature-wave Newtonian damping rate (the rate at which temperature perturbations are relaxed towards the mean state from which they deviate (Hartmann, 1981)). Whether the damping is enhanced or reduced depends on the spatial correlation of ozone and temperature perturbations. For example, when both ozone and temperature perturbations are positive, there is an increase in shortwave ozone heating due to its higher concentration, effectively reducing the damping of the temperature perturbation.

Albers and Nathan (2012) suggested two pathways through which ozone waves can affect the stratosphere. First, by affecting ozone advection through wave-ozone flux convergence, the zonal mean heating rate changes, consequently affecting the zonal mean temperature and wind. Second, the radiative effect of ozone waves impacts the temperature waves, and correspondingly the damping and propagation properties of planetary waves, and their EP-flux. Albers and Nathan (2012) further showed that the latter radiative effect reduces the planetary wave drag and modifies the wave amplitudes in a 1-dimensional Holton-Mass

model (Holton and Mass, 1976) coupled to a simplified ozone equation. These result in a colder upper stratosphere and a stronger polar vortex. In our paper we will focus on the second pathway - the direct radiative effect of ozone waves.

Several approaches have been used to asses the effect of ozone waves in GCMs. Some studies included climatological ozone waves in a constant or seasonally varying specified ozone field (Gabriel et al., 2007; Crook et al., 2008; Peters et al., 2015). While these studies found a significant effect for ozone waves, they do not include the dynamical interaction between the ozone

waves and other wave fields. A more direct approach is comparing two model simulations, one which includes ozone waves in the radiative transfer code and the other with only the zonally symmetric part of the ozone field passed onto the radiation code. These studies found that the runs which include ozone waves had a weaker, warmer northern winter polar vortex (Gillett et al., 2009; McCormack et al., 2009), stronger planetary wave drag, and a higher frequency of sudden stratospheric warmings (McCormack et al., 2009; Albers et al., 2013; Peters et al., 2015). However, there were some discrepancies regarding the timing

and strength of these effects. For example, McCormack et al. (2009) found the weakening of the polar vortex to occur in mid-Jan-Feb, while Gillett et al. (2009) found the weakening to occur earlier in Nov-Dec. We will discuss a possible explanation for this in the Summary.

A puzzling aspect of these results is the following. The radiative effects of ozone anomalies are expected to be strongest in summer and at lower latitudes, where solar radiation is strongest, while planetary waves, and in particular ozone and temper-

ature waves, are largest in winter and at high latitudes. Thus it is not clear how the radiative influence of ozone waves affects





the mid winter vortex – whether it is a seasonal amplification of an early fall radiative effect, or whether it is a mid-latitude amplification of radiative changes at the subtropical edge of the waves. For this we first need to determine the radiative effect of ozone waves on the wave temperature - to quantify the influence on thermal damping of temperature waves. This has not been explicitly examined using a CCM before. Also, given the possible involvement of the subtropics, we need to examine

the influence of the Quasi-Biennial Oscillation (QBO) - a tropical phenomenon in which the zonal mean zonal wind alternates from easterlies to westerlies, while the signal descends from the upper stratosphere with a mean period of approximately 28 months. An effect of the tropical flow on mid-latitudes is known to exist for the QBO, the phase of which is defined based on the direction of winds in the lower stratosphere. Interestingly, the QBO plays a role in communicating the effects of solar variations to high latitudes. Numerous studies have shown that the 11-year solar cycle correlates with the Arctic polar vortex

(e.g., Labitzke and Van Loon (1988); Labitzke et al. (2006); Matthes et al. (2010)). However, this signal correlates differently depending on the phase of the QBO: in the westerly phase solar maximum conditions correlate with a weak and warm polar vortex, while in the easterly QBO phase solar max conditions correlate with a stronger polar vortex. These studies suggest that the influence of radiative effects on the atmospheric circulation might depend on the phase of the QBO. It is thus plausible that the QBO modulates the effects of ozone waves at polar latitudes during winter as well. The QBO affects the propagation of

waves in the stratosphere resulting in a weaker and warmer winter polar vortex in the northern hemisphere during QBO east conditions (the Holton-Tan effect) (Holton and Tan, 1980). Several studies suggested a mechanism to explain this relationship. For example, Holton and Tan (1980) suggested that the poleward position of the subtropical zero wind line focuses the planetary wave activity to the polar vortex region during QBO east conditions, while Ruzmaikin et al. (2005) and Garfinkel et al. (2012) found that the subtropical meridional circulation of the QBO in the upper stratosphere is responsible for increased

EP-flux convergence in the polar vortex region. White et al. (2016) suggested the early winter planetary waves propagate differently and are more noninear under west QBO conditions. Gray et al. (2001) found that not only winds in the tropical lower stratosphere but also in the upper stratosphere influence the polar night jet. The Holton-Tan effect is found to be more robust in early winter (Holton and Tan, 1980), and we will see how this can be important to understand the influence of ozone planetary waves on the seasonal development of the winter polar vortex. To understand the mechanism through which ozone waves affect

the high latitude QBO signal we will take a synoptic approach, and analyze the life cycles of individual upward propagating wave events during fall, when the westerlies just get established in the stratosphere and planetary waves start propagating up from the troposphere. Besides illuminating the role or radiative ozone wave effects, this approach also provides a new look at how the tropical winds affect the polar vortex and the seasonal development of winter.

We will start by describing our model setup and output terms (Sec. 2). We will then show and quantify the direct radiative

ozone wave effects in terms of a modulation of the radiative damping (Sec. 3.1), and their corresponding influence on the atmospheric circulation (Sec 3.2). Section 3.3 will discuss the modulation of the seasonal cycle of the QBO and the Holton-Tan effect. Conclusions will be discussed in the last section. Radiative ozone wave effects during summer are discussed in the appendix.




## 2 Methodology

### 2.1 The WACCM Model

The model simulations were run with NCAR's CESM version 1.0.2, consisting of atmosphere (WACCM), ocean (POP), land
(CLM), and sea ice (CICE) components, based on the Community Climate System Model (CCSM4; Gent et al. (2011)). The

atmospheric component used for our experiments is the Whole Atmosphere Community Climate Model (WACCM) version 4
(Marsh et al., 2013) which has a horizontal resolution of 1.9°x 2.5° (latitude,longitude), 66 levels up to about 140 km, and
interactive chemistry (MOZART version 3). The chemistry module includes a total of 59 species, such as Ox, NOx, HOx,
ClOx, BrOx, and CH4, and 217 gas phase chemical reactions (Marsh et al., 2013). The model has a nudged Quasi-Biennial
Oscillation (QBO). The nudging is done by relaxation of the tropical zonal winds between 22S-22N, from 86 to 4 hPa towards

an averaged QBO cycle including a relaxation zone to the north and south. The QBO nudging is based on two idealized QBO-
east and QBO-west phases based on observational (rocketsonde) data, see further details in Matthes et al. (2010). Having a
QBO in the model is important for a realistic representation of the interaction between the tropical and extra-tropical region.
The solar cycle is prescribed as spectrally resolved daily variations following (Lean et al., 2005).

In our model experiments we kept greenhouse gases (GHGs) and ozone depleting substances (ODSs) fixed at 1960's con-

centration levels (pre ozone-hole) to get the cleanest signal possible for the ozone wave effects. Each experiment is a freely
running 100-year simulation (1955-2054) with interactive ocean and sea ice components. We run two 100-year simulations,
one using the full ozone field when calculating the radiative heating rates (hereafter 3DO3 run), and one using the zonally av-
eraged ozone field in the radiation code (hereafter ZMO3 run, see Table 1). In the ZMO3 run we use the full, zonally varying,
ozone field above 1hPa in the radiation code to avoid anomalous heating in the lower mesosphere due to the daily cycle (Gillett

et al., 2009). We transition from zonally averaged ozone to a full ozone field between 2hPa to 1hPa.

### 2.2 Diagnostics

We explicitly output the temperature time tendency terms from shortwave and longwave radiation, as well as from dynamics
and non-conservative processes. We use the explicit time tendency terms to evaluate the direct effect of ozone waves on the
temperature wave damping, and to compare it to other temperature time tendenfcy terms, in particular dynamics (see Figures

for details).

The radiative effects of ozone modulate the planetary waves, and so does their influence of the mean flow. These differences
add up to a difference in the climatological mean. We find that the effects of ozone waves are QBO dependent. To understand
the differences in planetary wave propagation depending on the phase of the QBO, and how ozone wave modulate them, we
look at the life cycles of individual events of upward wave propagation from the troposphere to the stratosphere. To do this,

first we calculate the daily $\overline{V'T'}$ at 85-45N, 100hPa, for both ZMO3 and 3DO3 runs. We then find, for each month, the 70th
percentile of the heat flux time series of both runs and select all the days for which the $\overline{V'T'}$ value exceeds this 70th percentile
threshold. We sort consecutive days into a single event, and events which are separated by less than 5 days are considered as
a single events. The central day of the event is considered as the day of the highest $\overline{V'T'}$ value. Then we classify the events





for east/west QBO according to the phase of the QBO using the monthly zonal mean zonal wind during the same month. The number of events for each month and model configuration is listed in Table 2. Similar results were found for higher $\overline{V'T'}$ thresholds, but the number of events was smaller. We will mostly examine the upward wave events in fall, during which there are no negative heat flux (downward wave coupling) events. The phase of the QBO is chosen using the zonal mean zonal wind at $50 - 30 hPa$, between $2.8S - 2.8N$ around the equator, where easterly (westerly) QBO winters are chosen where $u < -2.5 \frac{m}{sec}$ ($u > 5 \frac{m}{sec}$) during December.

The statistical significance of the differences between two model runs (e.g. east - west QBO or 3DO3 · ZMO3) is computed using a two-tailed t-test, with differences exceeding the 5% significance level marked by gray shading.

## 3 Results

### 3.1 The Direct Radiative Effect

Radiative effects of ozone waves modulate the radiative damping rate of temperature waves (see Appendix) in a way which depends on the spatial correlation between ozone and temperature waves. In the photochemically controlled upper stratosphere (above 10hPa) this correlation is negative, and in the transport controlled lower stratosphere (below 10hPa) the correlation is generally positive (Douglass et al., 1985a; Hartmann, 1981). The short-wave time tendencies of zonal wave 1 temperature amplitude is shown in Figure 1, alongside the wave 1 temperature and ozone amplitudes for reference, for northern hemisphere summer (Jun-Aug), fall (Sep-Nov), and winter (Dec-Feb). The tendencies were calculated using equation A2. The magnitude of the short-wave time tendency varies from $\pm 0.1 \frac{K}{day}$ to $\pm 0.2 \frac{K}{day}$, while the total tendency is about $\pm 0.5 \frac{K}{day}$ (not shown). It is generally positive in the lower stratosphere and negative in the upper stratosphere, with the zero line shifting from 5hPa in the tropical region to 2-3hPa at higher latitudes (Fig. 1a,1c,1e). The positive correlation at lower levels is due to the spatial correlation of ozone and temperature being positive in this region (not shown), as a result of ozone being dynamically controlled there (Douglass et al., 1985b). The negative tendency at upper levels is due to the negative correlation between ozone and temperature due to ozone being chemically controlled at high altitudes (Douglass et al., 1985b). As predicted by previous theoretical studies, we find that ozone wave radiative effects decrease (increase) the temperature wave damping where this correlation is positive (negative). This is true for zonal waves 2-4 as well (not shown). During summer, although the wave amplitudes are small (around 1K), the radiative effects coincide with the peak of the waves (Fig. 1b). This is also the case during fall, when the radiative effects are significant in the region where the temperature and ozone waves peak (around $7K$ and $7 \cdot 10^{-7} \frac{kg}{kg}$ respectively, $60 - 80N$, $10 - 1hPa$). To get a sense of the importance of the short-wave effect on temperature wave amplitudes, we explicitly calculate the ratio between this term and the corresponding time tendency due to long wave radiation (the radiative damping term, Figure 1, right column). We find that the shortwave tendency can reach up to 40% of the longwave tendency (Fig. 1d), however when the waves are stronger later in winter (around $16K$ and $10 \cdot 10^{-7} \frac{kg}{kg}$, $50 - 80N$, $10 - 1hPa$), the radiative effects at the region of peak wave amplitude are small (up to 10%, Figure 1f) as a result of radiation being weaker at higher latitudes during this period.



We further quantify total wave-amplitude weighted temperature time tendencies, for each calendar month separately, as follows:

$$\int\limits_{month} \frac{\int f(|T|) \cdot |T| dy dz}{\int |T| dy dz} dt$$

where $f(|T|) = \frac{d|T|_{tend1}}{d|T|_{tend2}}$, and tend1 and tend2 are different daily temperature wave 1 amplitude time tendency terms, averaged over 80-50N, 70-3mb. The ratios of different time tendency terms for each of the months Sep-Dec are shown in Table 3. We find that the relative shortwave contribution (columns 1-2) is strongest in fall (Sep-Oct) when there is enough radiation and the waves start to increase (about 19% of the longwave and 8% of the dynamical terms during October). By November this

contribution decreases by 50%, while the total radiative contribution increases compared to dynamics (3rd column) due to stronger decay of the wave through longwave radiation (4th column). We thus expect the direct ozone wave effect to have the strongest influence during Sep-Oct. In December, the dynamics play a larger relative role, indicating the waves are becoming more non-linear. We will show in Section 3.3 how these radiative effects in fall influence the differences in east/west QBO and modify the QBO signal at high latitudes and the mid winter polar vortex.

## 3.2   Radiative ozone wave effects on the atmospheric circulation

In this section we examine the differences in the circulation between the model run with full ozone fields passed to the radiation code (3DO3) and the run with the zonal mean ozone used for stratospheric heating rate calculations (ZMO3), as described is Section 2.1. The short-wave radiative forcing of temperature waves in the 3DO3 model run (Shown for wave 1 in Figure 1) constitutes the primary difference in wave forcing between the two runs. Thus we expect the 3DO3 run to have weaker

temperature wave damping in the lower to mid stratosphere, and stronger wave damping in the upper stratosphere.

The differences in the seasonal cycle of the polar cap temperature and the zonal mean zonal wind at mid-high latitudes (averaged over $55 - 75N$), between the 3DO3 and the ZMO3 runs are shown in Figure 2 (gray shading shows regions of statistical significance at 5% significance level). We see a significant effect in fall, when both the waves and radiation are strong enough (Section 3.1) and the vortex is established (green contours in Figure 2b). The polar night jet is stronger in

the lower stratosphere and weaker in the upper stratosphere in the 3DO3 run, with the upper stratospheric effect lasting until November (Fig. 2b). This is consistent with a weaker wave damping and thus stronger waves in the lower stratosphere, and stronger wave damping and thus weaker waves in the upper stratosphere (Fig. 1c). Correspondingly, the westerly jet in the lower stratosphere is stronger, and weaker in the upper stratosphere as a result of an upward shift of the wave absorption region (see next paragraph).

The above results suggest that the radiative effects of ozone waves in the 3DO3 run are most robustly established during Sep-Oct, (Fig. 2), when the winter vortex establishes, solar radiation reaches high latitudes, and the waves are strong enough to be radiatively affected, and weak enough for dynamics not to dominate completely. Under these conditions, the direct thermal damping of temperature waves by ozone waves has the largest influence. To understand how the ozone effects translate to dynamical changes, we examine the latitude-height structure of zonal wave 1 temperature and its shortwave radiative time

tendency, the zonal mean zonal wind and EP-Flux convergence, during September (Fig. 3). We find that the temperature



wave 1 amplitude is stronger throughout the stratosphere due to the weaker damping in the lower stratosphere (Fig. 3b), resulting in an upward displacement of the EP-flux convergence region where the waves decelerate the mean flow (Fig. 3c). Explicitly, there is decreased EP-flux convergence in the polar stratosphere, where the wave damping is reduced (note the gray lane marking the zero short wave radiative damping line), and increased EP-flux convergence in the upper stratosphere/lower

mesosphere where the wave damping is stronger, and at lower latitudes where more wave activity reaches due to the reduced high latitude convergence (Fig. 3a). This causes the polar night jet to strengthen in the lower stratosphere and weaken in the upper stratosphere, with a poleward tilt (Fig. 3d), while the latter lasts until November (Fig. 3d). The robust direct effects during September disappear after November (Fig. 2). This is most likely the result of the weak shortwave radiation at high latitudes and the dynamical effect taking over.

The results shown in Figure 2 appear to suggest that the winter mid latitude stratosphere is not sensitive to the inclusion of radiative ozone wave effects. While there is a significant radiative effect during fall, it seems to disappear later on. In the next section we will show, however, that this lack of a response is due to the response being oppositely signed between east and west QBO phases, so that there is a cancellation when all years are considered. A dependence of the mid-latitude response to various forcings on the phase of the QBO has been observed before in context of solar forcing, both for the 11-year solar cycle

(Labitzke and Van Loon, 1992) and for the 27-day period (Garfinkel et al., 2015). In the next section we will thus examine the response of the midlatitude QBO signal to the inclusion of ozone waves in the radiation code.

### 3.3 The modulation of the QBO signal

The influence of the tropical QBO phenomenon on the extra-tropical region results in a weaker and warmer polar night vortex during the easterly phase of the QBO, known as the Holton-Tan effect. Figure 4 shows the difference between the east and west

QBO phases of the seasonal evolution of the polar vortex and polar cap temperatures, overlain on the climatological seasonal cycle based on all years, for the 3DO3 and ZMO3 runs. In the 3DO3 run, the Holton-Tan effect starts in October, where the vortex is weaker (Fig. 4a) and warmer (Fig.4b) in the easterly QBO phase. In the ZMO3 run (Fig. 4c-4d), the Holton-Tan effect is delayed, with the robust signal starting about two months later, in January instead of November.

In order to understand the different seasonal development of the polar vortex, we will first examine the differences between

east and west QBO in the 3DO3 run when they just start, in October. For this we inspect the life cycle of upward propagating wave pulses entering the stratosphere (represented by 100mb positive heat flux events) and how they differ between east and west QBO. We take the strongest 30% of 100mb 80-45N mean heat flux events for a given month, and divide them according to the phase of the QBO. The time lag composites of anomalous $\overline{V'T'}$ (Fig. 5a) and zonal mean zonal wind averaged over the extratropical stratosphere (85-40N, 50-0.1mb, marked by the green rectangle in Figure 6a) for October events[1] show that while

the heat flux pulses are relatively similar in magnitude and length (Fig. 5a), the deceleration of the jet is different between the QBO phases. Specifically, during both QBO phases, the life cycle shows deceleration followed by acceleration but the deceleration is stronger during east QBO, and while in east QBO the acceleration is smaller than the initial deceleration, during west QBO the acceleration completely reverses it, leaving the vortex at similar strength. Since the anomalies are based on a

---

[1]We only show results for positive heat flux events since we did not find negative heat flux events during October.





climatology of the full run, we see part of the east-west QBO difference already at negative time lags, but this difference grows with each upward wave pulse. This is more clearly illustrated in latitude-height composites of the zonal mean zonal wind at different stages of the wave life cycle for east and west QBO phases (Figure 6). The tropical QBO signal is evident, as well as a small but significant midlatitude QBO signal of opposite signs. This midlatitude signal is evident between 40-60N at all

stages, even at negative time lags. During the peak of the event (days -3 to 3) we see a weakening of the zonal wind anomalies at high latitudes and all levels but this weakening is much clearer during east QBO. At later stages, on the other hand, the winds strengthen back, essentially spreading the initial anomaly between 40-60N to polar latitudes. The strengthening of the midlatitude QBO signal over the life cycle is seen clearly when looking at the differences between the east and west composites (Figure 7c). To isolate the effect of the wave pulse from the preexisting QBO signal, we composite the zonal mean zonal wind

time tendency (Figure 7). We see a clear deceleration of the vortex during the peak of the event (days 3 to -3) for both QBO phases with a slightly stronger deceleration during east QBO, but the largest difference is during the end of the life cycle (days 7 to 12)- while there is a very weak acceleration during east QBO, the acceleration is comparable in magnitude to the deceleration during west QBO.

To better understand the zonal mean wind we composite the momentum budget (see Andrews et al. (1987), Eq.3.5.2a)

(Figures 5c-5d). During east QBO events the deceleration is driven by a clear EP flux convergence which is counteracted by the Coriolis term, during west QBO, these terms are much weaker. This is quantified more clearly by time integrating the different time tendency terms over the life cycle (days $-10$ to day 20, values indicated in the figure legend). In particular, the time integrated $\frac{d\bar{U}}{dt}$ represents the reversibility of the wave life cycle. In the west QBO (Fig. 5d) events the positive value indicates the process is indeed more reversible, while the negative value in the east QBO composite (Fig. 5c) shows that a

significant part of the wave-induced deceleration of the mean flow remains after the life cycle has ended.

To understand why the life cycle is more reversible during west QBO events we look at the latitude-height daily time lag composits of EP-flux convergence anomalies (Fig, 8). There is stronger convergence at the high latitude upper stratosphere in the east QBO events at days -3 to 7 (Fig. 8a, 8c) while in the west QBO events there is increased convergence in the subtropical region (Fig 8b, 8c). This suggests the waves propagate up along the polar vortex and break in the upper polar stratosphere

during east QBO while they refract equatorwards in the middle stratosphere during the west QBO phase. This difference in wave propagation can be explained when examining the index of refraction composites before the wave pulses start (days -5 to -10, Fig. 9). The index of refraction is stronger in the high latitude upper stratosphere during east QBO, and stronger in the midlatitude subtropics during west QBO, consistent with the waves propagating to the upper polar stratosphere during east QBO and more equatorwards during west QBO. At later stages of the wave life cycle (days 8-17) there are significant EP-flux

divergence anomalies during west QBO, indicative of anomalous acceleration. This is consistent with a trailing-edge acceleration which is expected to occur under non-acceleration conditions which are satisfied when the waves are linear and damping is weak (Andrews et al., 1987). During east QBO, we see no such EP flux divergence region. This suggests the following picture: During fall, after the westerly winds get established and planetary waves start propagating up to the stratosphere, the waves are weak enough to be linear in the lower-mid stratosphere. Under these conditions, only waves which propagate up the polar

vortex to the upper stratosphere/mesosphere grow enough (due to the density effect) to break nonlinearly. This happens during



east QBO, and the deceleration induced by the breaking waves is irreversible in large part. During west QBO, the waves refract to the equator before reaching levels where they become significantly nonlinear, thus they decelerate the vortex when propagating up and accelerate it when refracting equatorwards. The strong acceleration is enabled due to non acceleration conditions being satisfied. The strong acceleration is enabled due to non acceleration conditions being satisfied [2]

To explicitly examine the degree to which non-acceleration conditions are satisfied, we inspect the enstrophy budget and see how the different terms balance during these heat flux events. Following Equation 3 from Smith (1983) we use:

$$\frac{\partial}{\partial t} \frac{\overline{q'^2}}{2} = -\overline{v'q'}\overline{q}_y - \overline{\frac{q'u'}{a cos\phi}\frac{\partial q'}{\partial \lambda}} - \overline{\frac{q'v'}{a}\frac{\partial q'}{\partial \phi}} + \overline{q'D'_{sw}} + \overline{q'D'_{lw}} - Resid \tag{1}$$

where

$$D' = \frac{Rf}{H\rho}\frac{\partial}{\partial z}\frac{\rho Q'}{N^2} \tag{2}$$

where $X'$ denotes deviation from the zonal mean and $q$ is the QG potential vorticity. On the right hand-side, the first term is the wave-mean flow interaction term, equivalent to the EP-flux divergence times the meridional gradient of the zonal mean potential vorticity ($\overline{q}_y$), the second and third terms are the non-linear terms, the fourth and fifth terms are the diabatic terms from shortwave and longwave radiation (where $Q'$ = temperature tendency from shortwave or longwave radiation) , and the last term is the residual of the total time tendency minus all the terms on the right hand side. Large nonlinear, damping and

residual terms indicate a violation of non-acceleration conditions (Andrews et al., 1987).

To avoid misinterpreting the differences in the enstrophy balance we normalize the events by the mean value of the heat flux amplitude entering the stratosphere at the peak of the events between day $-3$ and 3 ($\overline{V'T'}$ at 100mb). Figure 10 shows the time-lagged composites of the different enstrophy budget terms, averaged over 40-70N, 50-1mb. The averaging area was chosen based on examination of latitude-height composites. As expected, the nonlinear terms (red line in Fig. 10a) are larger

during east QBO, while during west QBO events the wave transience term is more significant (black line in Fig. 10b). The results are consistent with White et al. (2016) who used reanalysis data to study the different seasonal cycles between east/west QBO, and found that non-linear interactions are stronger for east QBO years during Nov-Jan.

To understand the role of ozone waves we repeat the analysis shown in Figure 8 for the ZMO3 run. The results are shown in Figure 11. The main point to note is the lack of strong positive anomalies at positive time lags during west QBO, suggesting

the wave induced deceleration is not as reversible as in the 3DO3 run. This weaker trailing-edge deceleration for the ZMO3 run is consistent with there being stronger radiative damping of the waves in the lower-mid stratosphere. In addition, we see weaker EP flux divergence compared to the 3DO3 run for east QBO at days 4 to 7, consistent with ozone waves increasing the upper stratospheric wave damping.

---

[2]Strictly speaking, the non acceleration conditions apply for the wave activity equation (the enstrophy equation divided by the PV gradient and density, so we are assuming the PV gradient is not zero over the domain and time periods we are examining. Also, non acceleration conditions apply for a statistical steady state. Here we are interested in the net deceleration over the wave life cycle, and can assume quite safely that the time averaged (over the wave life cycle) enstrophy time tendency vanishes over the wave life cycle.





The above results suggest that ozone waves affect the EP flux divergence in opposite manners during east and west QBO phases. A closer examination of the EP flux in our runs shows the vertical EP flux is strongly converging while the meridional EP flux is strongly diverging (not shown). Figure 12 shows the difference between 3DO3 and ZMO3 runs of the latitude-height composites during the later stages of the wave life cycle (days 4 to 7 and for days 8 to 11), for east (Fig. 12a) and west (Fig. 12b) QBO. We find that both the vertical convergence and the meridional divergence are stronger in the 3DO3 run compared to the ZMO3 run (not shown), but while during east QBO phase the vertical convergence on days 4 to 7 dominates the EP flux divergence (Fig. 12a), during west QBO phase, the meridional convergence on days 8 to 11 dominates the EP flux divergence (Fig. 12b). This results in more EP flux convergence in the 3DO3 run during east QBO and more EP flux divergence in the 3DO3 run during west QBO. Correspondingly the vortex is weaker in the 3DO3 run during east QBO and stronger during west QBO.

The results shown so far were for October. The differences in individual life cycles lead to a slightly stronger deceleration and warming of the polar vortex during east QBO compared to west QBO phases. Similar differences are also found in November. These differences between east and west QBO cycles continue and are thus further intensified later in November (not shown), so that by early winter the polar vortex is weaker during east QBO years compared to WQBO years (Fig. 4a). We also find a stronger poleward meridional circulation in the subtropical lower stratosphere, which extends to higher levels and latitudes at positive time lags (not shown). These results are consistent with Garfinkel et al. (2012) though they used a different model configuration (WACCM version 3.1 with fixed SSTs and perpetual winter radiative forcing, compared to our freely running model with interactive ocean and sea ice components version).

The radiative effects of ozone waves, which are strongest in fall (Table 3, 1st and second column) cause the stronger vertical and meridional convergence, which dominate the EP-flux convergence differently under the different QBO phases and cause the stronger deceleration in the east QBO phase and stronger recovery in the west QBO phase, resulting in the earlier Holton-Tan effect seen in the 3DO3 run. Kodera and Kuroda (2002) found that solar cycle variations have the largest influence on stratospheric winds in the transition period between the time when the stratosphere is radiatively controlled (early winter) and the time when it is dynamically controlled by wave-mean flow interactions (late winter). Interestingly, we also find that ozone wave effects, which one might consider to be another mechanism through which anomalies in solar heating affect the stratosphere, have the largest impact during the transition period which occurs in our model in November. The transition period was identified by examination of the zonal mean temperature time tendency, which was either controlled by the tendency from radiation or dynamics (not shown).

Finally, we look at the seasonal differences in the development between the 3DO3 and ZMO3 runs at each QBO phase separately, where the phase of the QBO for the entire winter season is defined in October. Figure 13 shown the daily climatology of the EP-flux divergence in the 3DO3 (black) and ZMO3 (blue) runs and the difference (red), and the zonal mean zonal wind differences (green) in east (Fig. 13a) and west (Fig. 13b) QBO years, averaged over 85-45N and 10-0.1hPa. As seen in section 3.1, influence of the direct ozone wave effect starts with increased EP-flux convergence in the upper stratosphere during September (Fig. 3c). The increased convergence strengthens and descends lower down resulting in a weaker polar vortex by November during east QBO years (Fig. 13a, green line). This initiates a polar night jet oscillation in mid winter





due to a preconditioning of the vortex: less upward wave propagation and deceleration (in Nov-Dec), followed by a stronger vortex from mid-December, although the signal is not statistically significant due to the noisy winter resulting of inter-annual variability (Fig. 13a, red and green lines). In addition, we find differences in the zonal mean ozone concentrations in the polar region starting in September of around 6-8% in the high latitude mid stratosphere (not shown). We note that in a previous study

by Albers et al. (2013) it was mentioned that the zonal mean ozone variations were negligible. In our study, we do find these small differences to be statistically significant, and inspecting the zonal mean tendency from shortwave heating show these effects can reach up to 10% of the climatological time tendency in early winter ($0.05\frac{K}{day}$ in Sep-Oct), however, they are much weaker later in mid winter (not shown). These changes can also invite additional feedbacks on the ozone wave radiative effects through modulation of the ozone wave amplitudes.

In the west QBO years, the apparent opposite effect of the ozone waves on the EP-flux convergence is due to the reversible life-cycle of upward propagating wave events starting earlier in winter which allows a stronger vortex by December in the 3DO3 run (Fig. 13b, red and green lines). The latter allows enhanced wave propagation (also accompanied by overall stronger wave pulses entering the stratosphere) and then deceleration of the flow in January (Fig. 13b). These induced changes in the circulation cause a dynamical cooling (heating) during December (January) in the lower stratosphere, and heating (cooling)

during December (January) in the upper stratosphere/lower mesosphere (not shown). When considering all 100 years of the model runs, the response to ozone wave radiative effects in Dec-Jan is missing (see the lack of statistically significant signal during these months in Figure 2) as a result of them being of opposite sign during these months in east and west QBO phases (Fig. 13, green line).

## 4 Conclusions

The direct radiative effect of ozone waves is studied using the WACCM model. We find these effects can be important in early winter when the polar night jet is formed and there is still enough radiation where the ozone and temperature waves are relatively strong. As a result there is an increase in the mean flow deceleration from wave absorption in the upper stratosphere in the 3DO3 run, while decreasing it in the lower stratosphere (September, see Figure 3c). The acceleration/deceleration pattern has a poleward tilt, effectively confining the polar winter jet to higher latitudes (Fig. 3d). This helps to focus the waves to the

higher latitudes and as a result a stronger deceleration of the winds (McIntyre, 1982) in the following months. This happens due to weaker wave damping on temperature waves in the lower stratosphere and stronger damping in the upper stratosphere, as expected by earlier theoretical studies. These effects accumulate to affect the development of the mid winter polar jet. While we find the zonal mean winds to weaken until November, (McCormack et al., 2009) found a response in mid winter: mid-Jan to February. However, their experiment was run for Dec-Mar with similar initial conditions. Our results are consistent with

those of Gillett et al. (2009) who found the weakening to occur earlier in October to mid-December, and they used a 40-year simulation of the entire seasonal cycle, and their model has a realistic QBO which is spontaneously produced (Scinocca et al., 2008), thus having more resemblance to our model setup. A possible explanation to these discrepancies between McCormack et al. (2009) to Gillett et al. (2009) and our results is the inclusion of the full seasonal cycle, which makes the differences appear




in early winter and the late winter signal then becomes too noisy, while in starting the simulation in mid winter (Dec), when radiation is still weak at high latitudes, with similar initial conditions helps to get a cleaner signal in late winter. The existence of a realistic QBO can also explain this as we saw the apparent opposite effect of ozone waves on east/west QBO can mask the signal in mid-winter. It can be insightful to repeat the analysis of Gillett et al. (2009) considering the phase of the QBO and see

how our results apply to other climate models.

We further found that the effects of ozone waves depend on the phase of the QBO in early winter. A synoptic analysis of east/west QBO differences in early winter is used to understand the mechanism to explain this. The events of upward wave propagation behave differently. The life cycle of the west QBO events is more reversible in early winter (Fig. 5d), and the polar night jet can recover after upward propagating wave events (Fig. 5b). These differences in the individual events add up, and

the cumulative effect is consistent with the known Holton-Tan effect resulting in a stronger polar vortex in west QBO years, which occurs a month earlier in the 3DO3 run (Fig. 4). In the east QBO events there is stronger EP-flux convergence at the upper levels (Fig 8a), which is further increased in the 3DO3 run in early winter, and as winter progresses the deceleration is extended poleward and downward. The ozone waves increase both meridional and vertical convergence of the EP-flux in early winter (Fig. 12), however, they are opposite in sign at higher latitudes, and in the west QBO phase the meridional part is

more dominant at positive time lags. This causes the weaker deceleration at the trailing edge of the waves responsible for the reversibility of the west QBO events to appears a month earlier in the 3DO3 run, and resulting in the earlier Holton-Tan signal. Our model setup used fixed GHGs and ODSs at 1960's levels, where ozone waves are weaker compared to the 1990's (not shown). It is possible that the ozone wave effects found in this study will be much stronger under climate change conditions and will have a larger impact on the inter-annual variability.

The analysis provided for the life cycle of wave events at different QBO phases provides an additional mechanism to understand the Holton-Tan effect. In particular, the influence of ozone wave effects might explain the weak Holton-Tan effect produced by climate models forecasts (Smith et al., 2016), and might improve the predictability if included (Scaife et al., 2014).

Previous studies showed the relationship between solar cycle effects through modulation of the tropical stratospheric ozone heating affect the high latitude polar vortex depending on the phase of the QBO (eg. Labitzke and Van Loon (1988), Garfinkel

et al. (2015)). Our result provide an additional mechanism through which solar cycle effects might play a role in the circulation of the stratosphere (through modulating the direct ozone wave effects on the temperature waves) and their sensitivity to the tropical stratospheric winds, and requires further study.





## 5 Tables

**Table 1.** Model setup for 3DO3 and ZMO3 experiments

| Experiment | QBO | SST/Sea Ice | Ozone passed to radiation code |
|---|---|---|---|
| 3DO3 | nudged | interactive | Full field |
| ZMO3 | nudged | interactive | Zonally averaged |

**Table 2.** Number of positive heat flux events for east/west QBO phase for Oct-Dec for the 3DO3 and the ZMO3 experiments.

| Month | EQBO(3D) | WQBO(3D) | EQBO(ZM) | WQBO(ZM) |
|---|---|---|---|---|
| Oct | 55 | 46 | 44 | 43 |
| Nov | 52 | 48 | 44 | 35 |
| Dec | 52 | 39 | 47 | 38 |

**Table 3.** Showing the seasonal development (Oct-Dec) of the integrated values of the following: $\frac{\int f(|T|)\cdot|T|dydz}{\int|T|dydz}$, where $f(|T|) = \frac{d|T|_{tend1}}{d|T|_{tend2}}$, and tend1 and tend2 are the temperature wave 1 amplitude tendencies from short-wave/long-wave radiation and from dynamics, averaged over 80-40N, 50-0.5mb, for the 3DO3 run.

| Month | $\frac{swr}{lwr}$ | $\frac{swr}{dyn}$ | $\frac{rad}{dyn}$ | $\frac{lwr}{dyn}$ |
|---|---|---|---|---|
| Sep | 0.1438 | 0.0813 | 0.1552 | 0.1865 |
| Oct | 0.1897 | 0.0882 | 0.3918 | 0.4458 |
| Nov | 0.0997 | 0.0481 | 0.4598 | 0.4875 |
| Dec | 0.0663 | 0.0286 | 0.4407 | 0.4532 |



# 6 Figures





(a) Jun-Aug

(b) Jun-Aug

(c) Sep-Nov

(d) Sep-Nov

(e) Dec-Feb

(f) Dec-Feb

**Figure 1.** Monthly mean temperature tendency from SWR of temperature zonal wave 1 amplitude (left), $\%\frac{SWR}{LWR}$ (fraction of the tendency from SWR of temperature zonal wave 1 amplitude compared to LWR (right), in the northern hemisphere during Jun-Aug (top), Oct-Nov (mid) and Dec-Feb (bot). Temperature (ozone) wave 1 amplitude in K ($10^{-7}\frac{kg}{kg}$) are shown in gray (green) contours.





(a) Tzm (90-60N) 3D-ZM

(b) Uzm (75-55N) 3D-ZM

**Figure 2.** Height-time differences between the 3DO3 and ZMO3 run for all years for zonal mean temperature, zonal wind, EP-flux divergence, and temperature zonal wave 1 amplitude (from top to bottom). The difference between the 3DO3 and the ZMO3 model runs are indicated by the colored contours, the climatology of the 3DO3 run is shown by the green contours. Statistically significant areas are shown by gray shading.



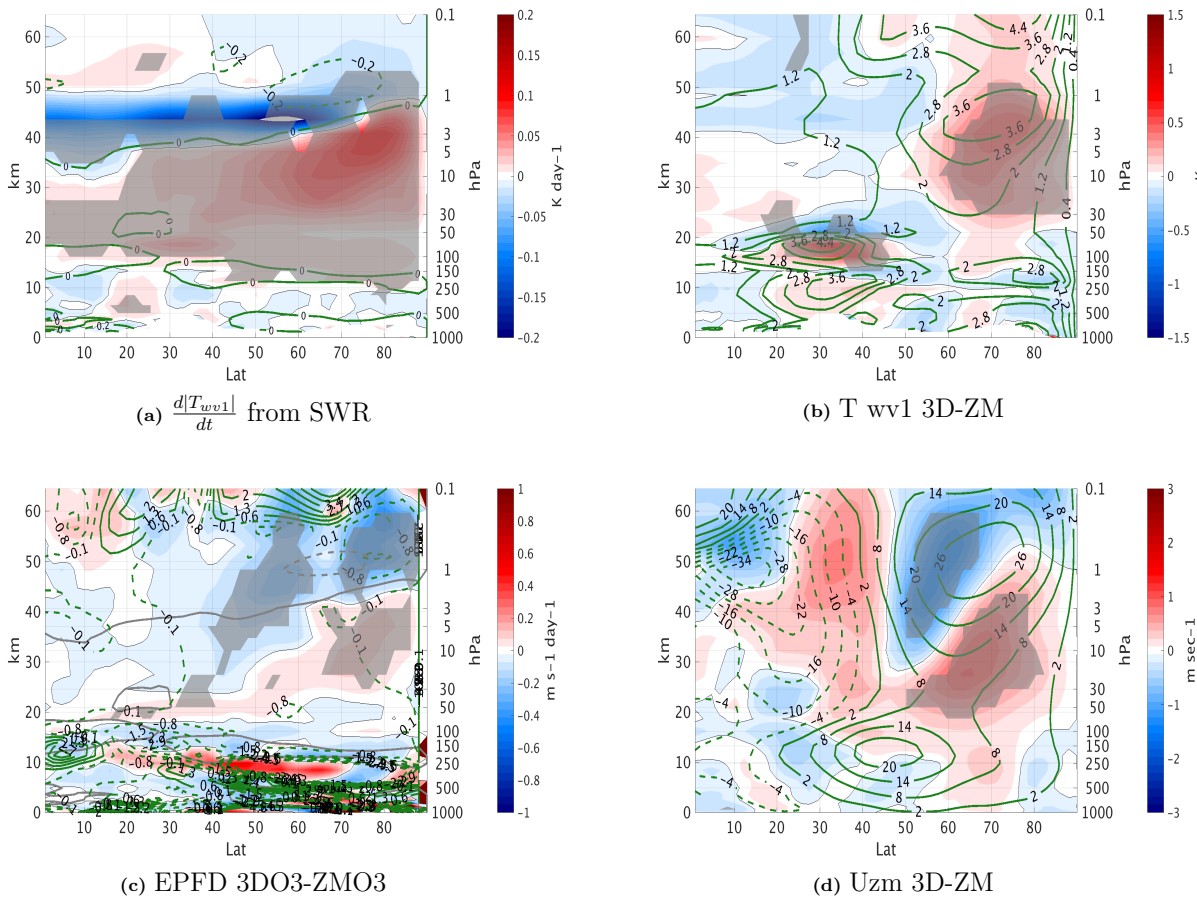

(a) $\frac{d|T_{wv1}|}{dt}$ from SWR

(b) T wv1 3D-ZM

(c) EPFD 3DO3-ZMO3

(d) Uzm 3D-ZM

**Figure 3.** September mean differences between the 3DO3 and ZMO3 run for all years for temperature wave 1 amplitude tendency from short-wave radiation (3a), temperature zonal wave 1 amplitude (3b), EP-flux divergence (3c), and zonal wind (3d) . In Figure 3c the gray line in the upper stratosphere indicated the height where ozone and temperature zonal wave 1 correlation change from positive to negative. Statistically significant areas are shown by gray shading.



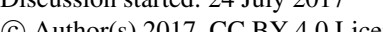

(a) Uzm EQBO-WQBO 3DO3

(b) Tzm EQBO-WQBO 3DO3

(c) Uzm EQBO-WQBO ZMO3

(d) Tzm EQBO-WQBO ZMO3

**Figure 4.** Daily climatology differences between east and west QBO phase of the zonal mean zonal mean zonal wind averaged over 75-55N for the 3DO3 (4a) and ZMO3 (4c) runs, and the zonal mean temperature averaged over 90-66N for the 3DO3 (4b) and ZMO3 (4d) runs, for Sep-Mar. Statistically significant areas are shown by gray shading.





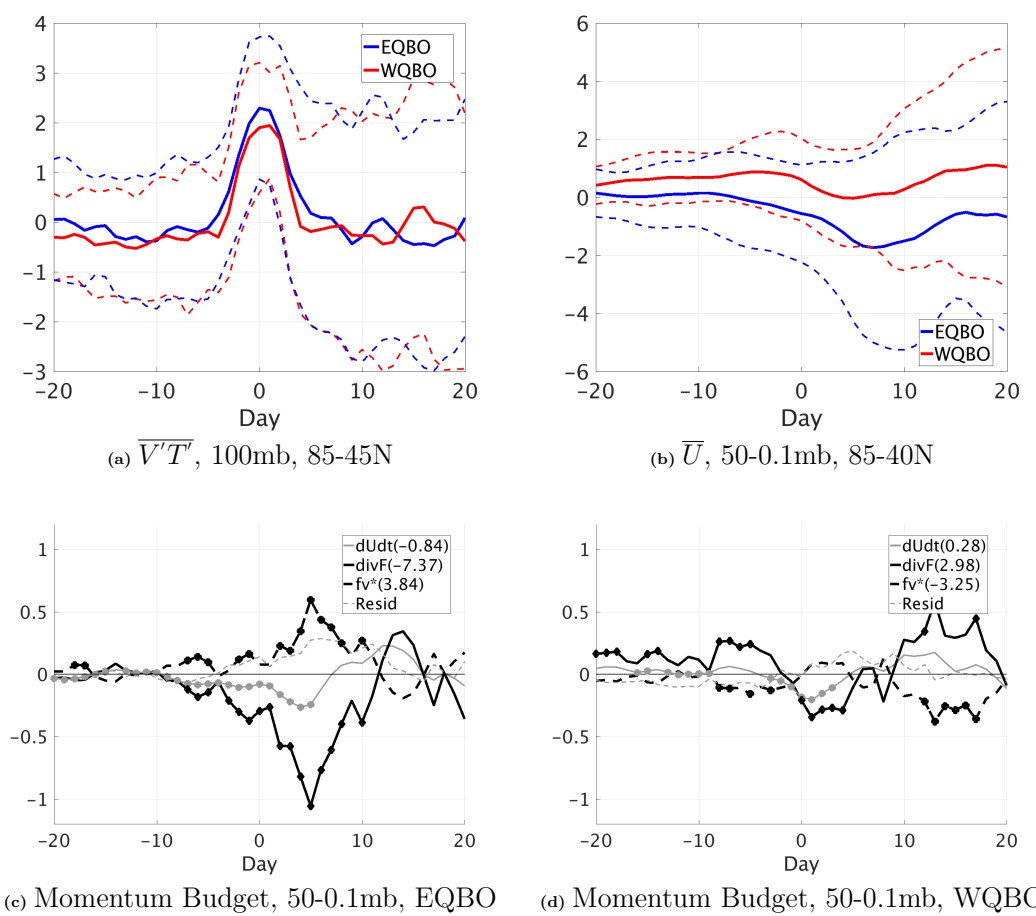

**Figure 5.** Time lag composites for the upward wave pulse events during October in the 3DO3 run. (a) $\overline{V'T'}$ averaged over 85-45N at 100mb. (b-d) The extratropical stratospheric averages (50-0.1mb, 85-40N, marked by the green rectangle in 6a))of: (b) $\overline{U}$, dashed lines show $\pm 1$ standard deviation. (c-d) Momentum budget terms for east and west QBO events respectively. Shown are the total time tendency (thin gray), $f\overline{v}^*$ (dashed black) and the residual (gray dashed) with their integrated value from day -10 to 20 denoted in the figure legend.



(a) $\overline{U}$, EQBO, 3DO3

(b) $\overline{U}$, WQBO, 3DO3

(c) $\overline{U}$, (E-W)QBO, 3DO3

**Figure 6.** Time lag composit of the zonal mean zonal wind anomalies for east QBO (6a), west QBO (6b), and the difference between them (6c), for the positive heat flux events from the 3DO3 run of October. The green box in Figure 6a shows the area of averaging for Figures 5b-5d.

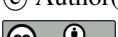



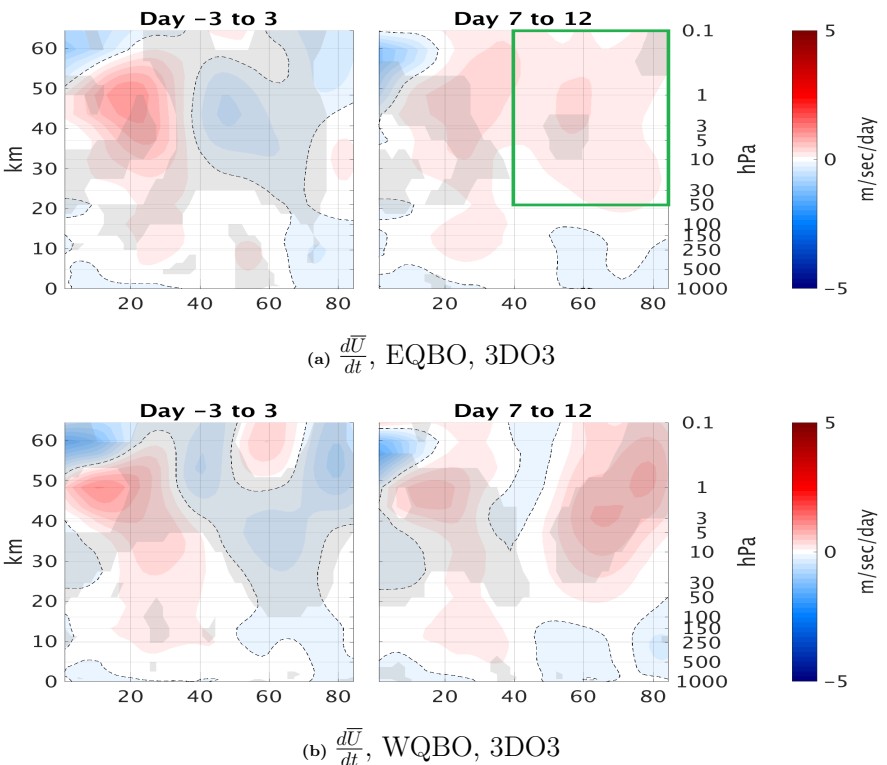

**Figure 7.** Time lag composit of the zonal mean zonal wind time tendency for east QBO (7a) and west QBO (7b), for the positive heat flux events from the 3DO3 run of October. The green box in Figure 7a shows the area of averaging for Figures 5b-5d.



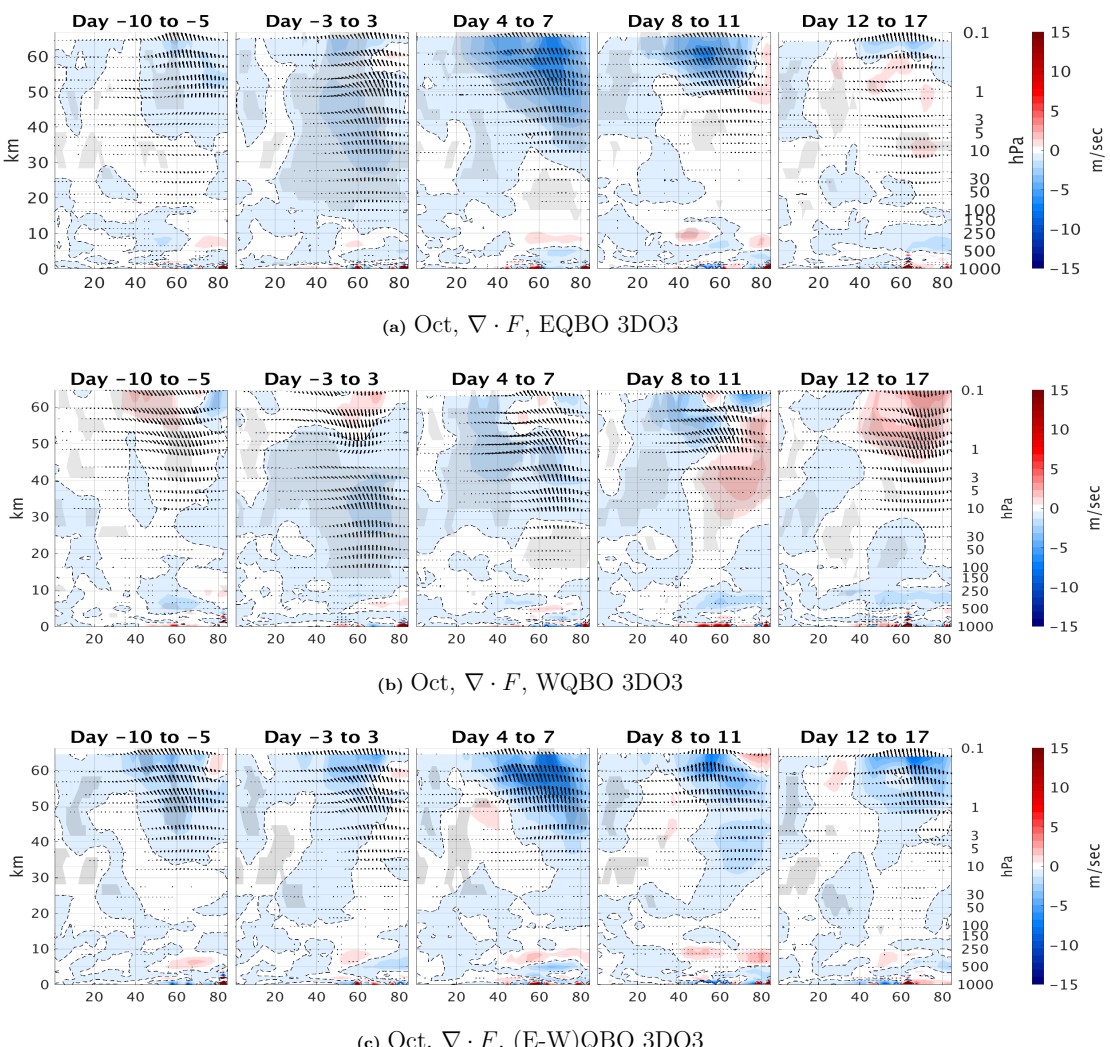

(a) Oct, $\nabla \cdot F$, EQBO 3DO3

(b) Oct, $\nabla \cdot F$, WQBO 3DO3

(c) Oct, $\nabla \cdot F$, (E-W)QBO 3DO3

**Figure 8.** Lat-height time lag composits of EP-flux divergence (anomalies from the climatology) for the positive heat flux events (70th percentile of $\overline{V'T'}$ at 100mb 85-45N), for east (8a), west (8b) and the their differences (8c) for October events for the 3DO3 run. Statistically significant areas are shown by gray shading.




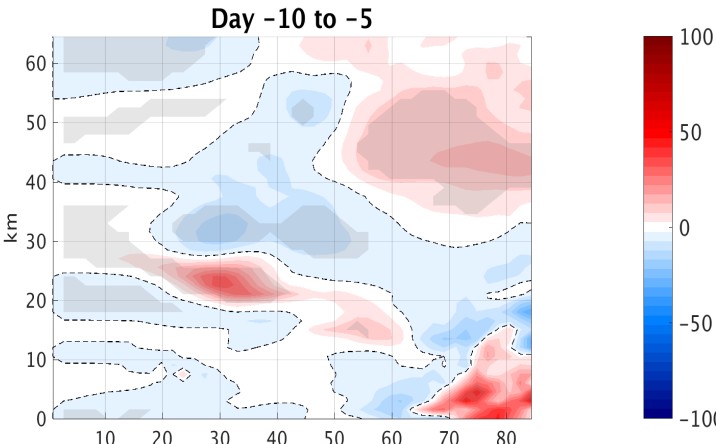

**Figure 9.** Index of refraction ($n^2 = \frac{N^2}{f_o^2}\frac{\overline{q}_y}{\overline{U}-c} - k^2\frac{N^2}{f^2} + F(N^2)\frac{N^2}{f^2}$, see eq.5,6 in Harnik and Lindzen (2001))at days $-10$ to $-5$ for the difference between east and west QBO in the 3DO3 run.



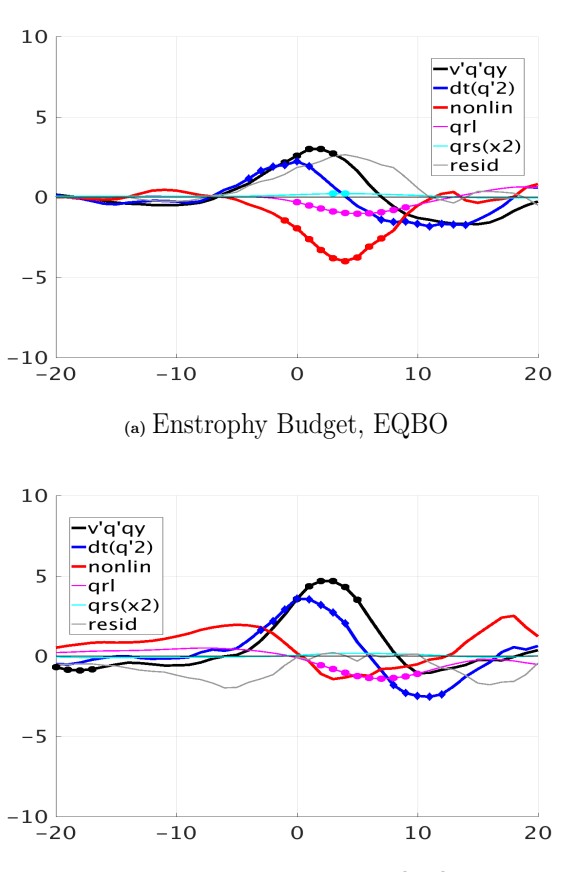

(a) Enstrophy Budget, EQBO

(b) Enstrophy Budget, WQBO

**Figure 10.** Time lag composit of the enstrophy budget terms normalized by the wave-mean flow term $\overline{v'q'}\overline{q}_y$ at days $-3$ to $3$ for east (top), and west (bot) QBO, averaged over 70-40N, 50-0.1mb, for the positive heat flux events from the 3DO3 run of October.





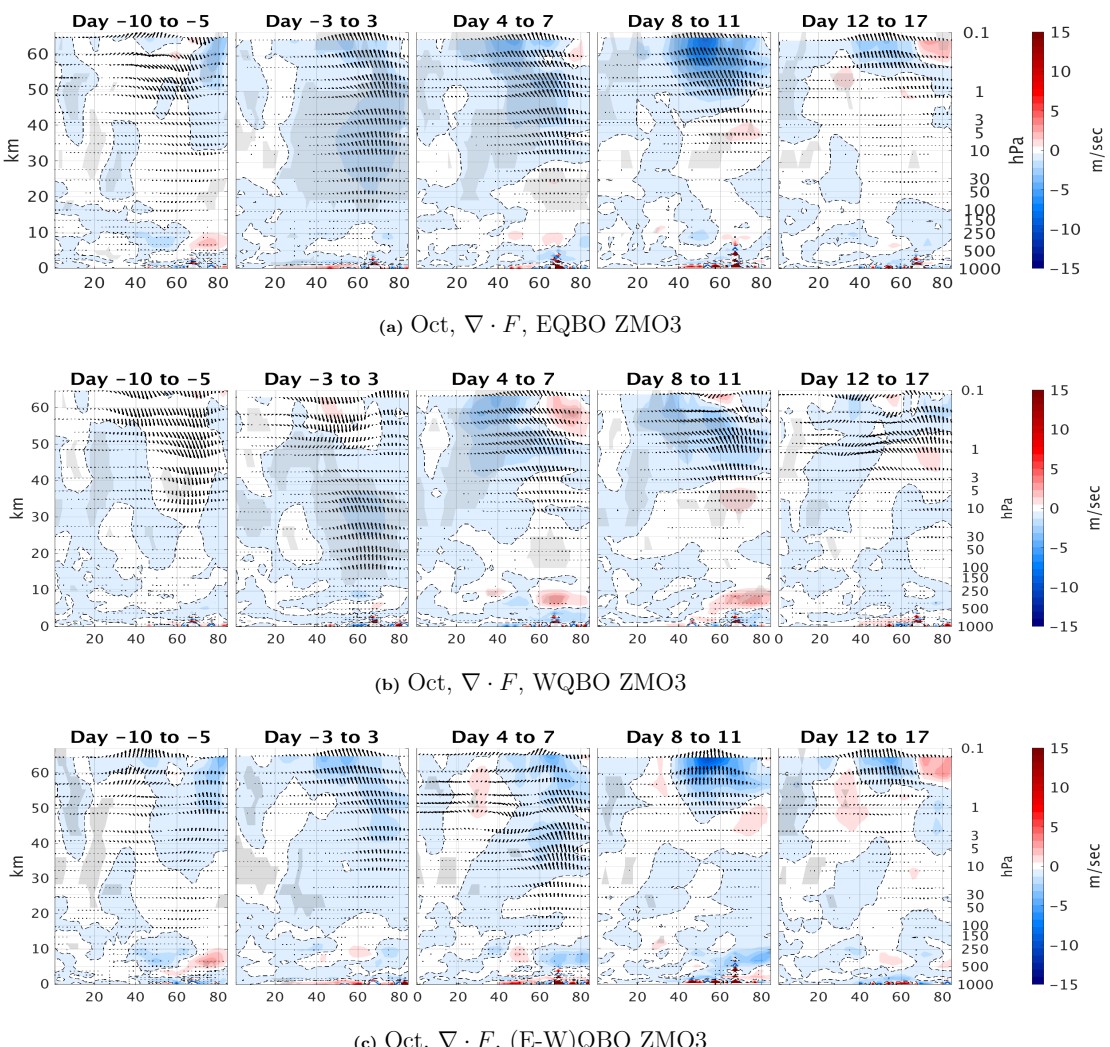

**Figure 11.** Lat-height time lag composits of EP-flux divergence anomalies from the climatology) for the positive heat flux EQBO (top), WQBO (min), and the difference between them (bot), for October events (70th percentile of $\overline{V'T'}$ at 100mb 85-45N) of the ZMO3 run. Statistically significant areas are shown by gray shading.





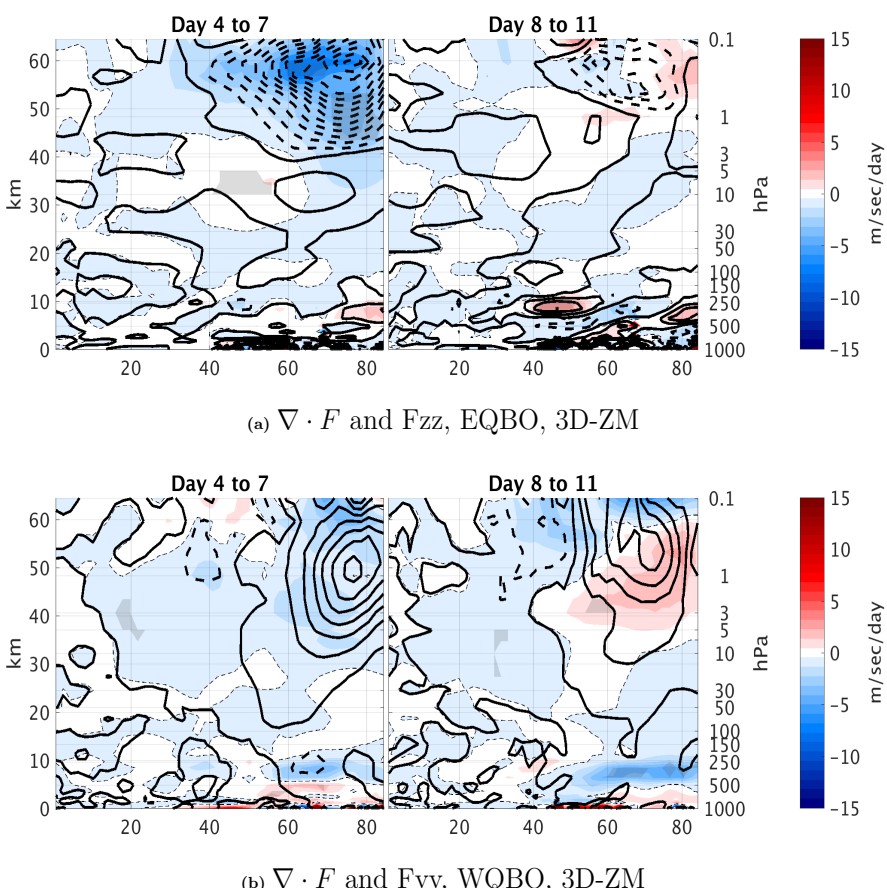

**Figure 12.** Lat-height time lag composits differences of the 3DO3 and ZMO3 runs of the EP-flux divergence (colors) and Fzz (contours) for east QBO (12a), and the EP-flux divergence (colors) and Fyy (contours) for west QBO (12b), for October events (70th percentile of $\overline{V'T'}$ at 100mb 85-45N). Dashed contours indicate negative values. Statistically significant areas are shown by gray shading.



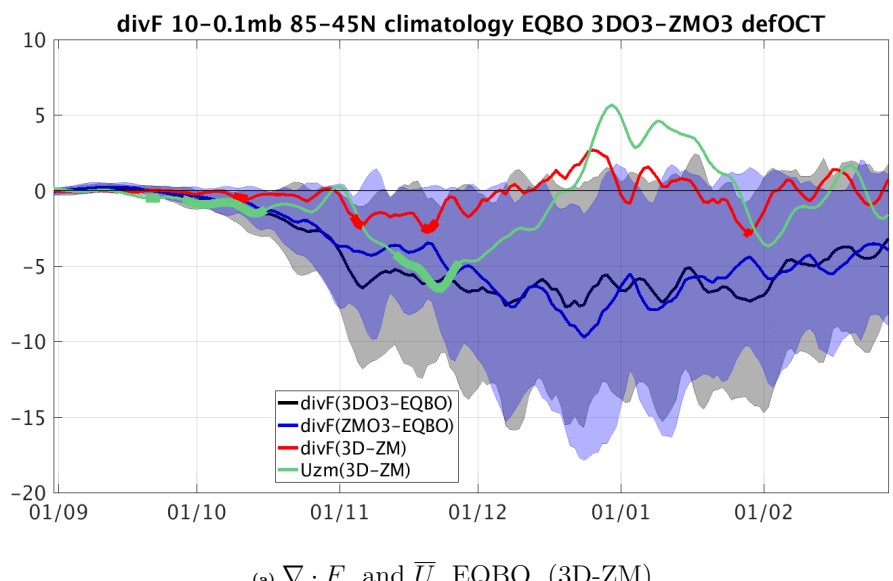

(a) $\nabla \cdot F$, and $\overline{U}$, EQBO, (3D-ZM)

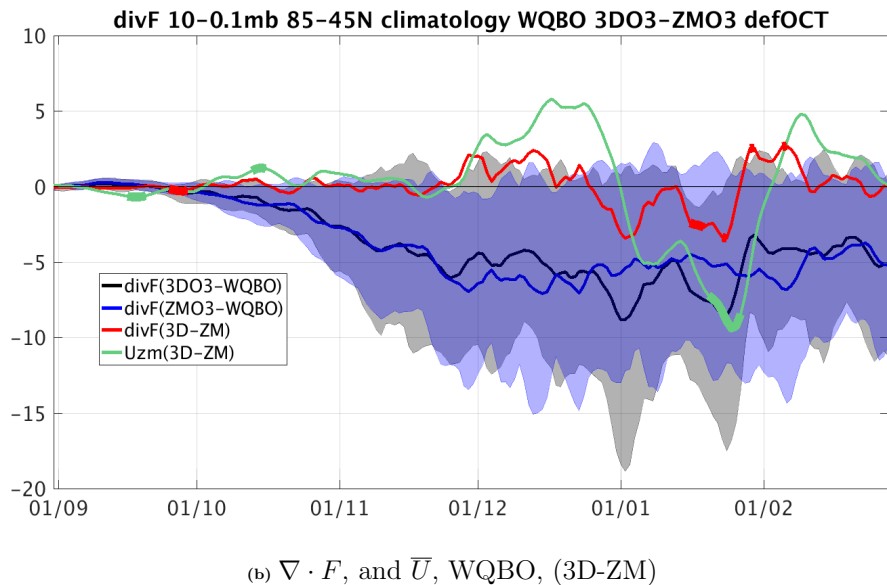

(b) $\nabla \cdot F$, and $\overline{U}$, WQBO, (3D-ZM)

**Figure 13.** Daily climatology of EQBO (top) and WQBO (bot) years (defined by October) averaged over 10-0.1mb, 85-45N. The EP-flux divergence for 3DO3 run (black), for ZMO3 run (blue) and their difference (red), with the difference between 3D and ZM runs of the zonal mean zonal wind in green. Gray and Blue shading indicated $\pm 1$ standard deviation from the mean of the 3DO3 and ZMO3 runs correspondingly.




## Appendix A: Appendix

### A1 Estimating the direct ozone effect (wave 1 amplitude tendencies)

We focus on zonal wave number 1 since it is the most dominant in the stratosphere. The main balance of temperature time tendency is given by:

$$\frac{dT}{dt} = \frac{dT}{dt}_{dynamics} + \frac{dT}{dt}_{shortwave} + \frac{dT}{dt}_{longwave} \tag{A1}$$

For the zonal wavenumber 1 amplitude balance we use the equations above, apply Fourier transform and take the first wave component. After that we have the following complex terms for temperature wave balance ($s1$ denoting first Fourier component):

$$\widetilde{\frac{dT}{dt}}^{s1} = \widetilde{\frac{dT}{dt}}_{dynamics}^{s1} + \widetilde{\frac{dT}{dt}}_{shortwave}^{s1} + \widetilde{\frac{dT}{dt}}_{longwave}^{s1} \tag{A2}$$

To estimate the time tendency tendency of the temperature wave amplitude from each term in each time step we use the following procedure:

1. Calculate the complex of the next time step from each term: $\widetilde{X}_{term}^{j+1} = \widetilde{X}^j + \widetilde{\frac{dT}{dt}}_{term}^{j}$, where "term" is either advection (total or one component), residual, or each of the tendencies from the model/reanalysis.

2. Calculate the change in amplitude: $D_{term}^j = |\widetilde{X}_{term}^{j+1}| - |\widetilde{X}^j|$, where $D_{term}^j$ is the amplitude tendency from a specific term.

It is important to note that this calculation implies the amplitude tendencies from each term do not add up to the total time tendency, however it represents best how each process "attempts" to the change the wave amplitude.

### A2 Radiative ozone wave effects on the atmospheric circulation during Summer

In sections 3.1-3.2 we showed the direct radiative effect of ozone waves on the circulation during September. Here we examine the differences between 3DO3 and ZMO3 runs during summer to verify that the September anomalies are not simply carried over from Summer. In particular, an examination of the 3DO3 minus ZMO3 zonal mean short wave heating during summer (Fig. A1c) reveals a thin band of stronger heating in the 3DO3 run, right at the levels where the model changes back to using 3D ozone in the radiation code in the upper stratosphere which persists into fall. Though this region is significantly reduced to a very small latitude range in early winter (less than 5 degrees in the subtropical region), we need to verify that it is not the source of differences between the 3DO3 and ZMO3 fields during fall and winter.

We find a few indications that this is not the case. First, looking at the zonal mean temperature, and the contribution of dynamics to the temperature time tendency, we find small but significant differences in the zonal mean temperature (Fig. A1a). The polar stratosphere is warmer above 20hPa and colder below in the 3DO3 run during May-Aug by about $1K$. Similar





differences are found in Gillett et al. (2009) (Fig. 3d). These differences are dynamically driven as indicated by the zonal mean temperature time tendency from dynamics (Fig. A1b). It is possible however, that the source of differences in the dynamical time tendencies is this anomalous band of short wave heating. Fig. A2) shows the 3DO3-ZMO3 differences of different terms in the zonal mean zonal wind time tendency equation. The zonal mean zonal wind of the 3DO3 run is more westerly in the

subtropical lower stratosphere in July, extending upward and poleward until August (Fig. A2a). There is a vertical displacement of the EP-flux convergence height, with decreased convergence in the lower stratosphere and increased convergence above 30mb (Fig. A2b), well below the region of negative ozone-temperature correlation (indicated by the gray line in the figures). This demonstrates that the vertical displacement of the convergence region is due to the waves reaching higher due to their stronger amplitudes. The total time tendency and the related zonal mean zonal wind anomalies are governed by these changes

only during Aug-Sep. Earlier in summer, the time tendency is controlled by the tendency from the Coriolis torque term ($f\overline{v}^*$) above 30mb (Fig. A2c) and by the EP-flux convergence below.

Finally, in addition to the runs described in this paper, we conducted four 40-year time slice experiments, for which we specified constant east or west QBO phases, for 3DO3 and ZMO3. While the summer heating bands also appeared during summer in these runs, the differences in the Holton-Tan effect between 3DO3 and ZMO3 runs during fall and winter were not

found. An examination of October upward wave pulses showed that in both runs there is a stronger EP flux divergence during the late stages of the wave life cycles in west compared to east QBO phases, but this acceleration is due to nonlinear wave-mean flow interactions rather than to a linear trailing edge acceleration. Correspondingly, the waves are stronger at 100mb in the time slice experiments during October (we are still examining the reasons for these differences). Nonetheless, this suggests that summer heating band is not the source of differences between the 3DO3 and ZMO3 runs found in fall and winter in our

time-varying QBO 100-year experiments.





**Figure A1.** Monthly climatology differences between 3D and ZM ozone runs during summer, Jun-Sep of the zonal mean temperature (A1a), zonal mean temperature tendency from dynamics (A1b), and short-wave radiation (A1c). Statistically significant areas are shown by gray shading.





(a) $\overline{U}$, 3D-ZM

(b) $\nabla \cdot F$, 3D-ZM

(c) $f\overline{v}^*$, 3D-ZM

(d) $\frac{d\overline{U}}{dt}$ 3D-ZM

**Figure A2.** Monthly climatology differences between 3D and ZM ozone runs during summer, Jun-Sep of the zonal mean zonal wind (A2a), zonal mean zonal wind tendency from EP-flux convergence (A2b), the time ternency from the Coriolis term (A2c), and the total time tendency (A2d). Statistically significant areas are shown by gray shading.

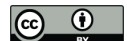



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
