# Peer review of "Radiative effects of ozone waves on the Northern Hemisphere polar vortex and its modulation by the QBO"

_Atmospheric Chemistry and Physics, 2017_

## Referee Comment (RC1) · Anonymous Referee #1 · 6 Sep 2017

Atmospheric and Chemistry and Physics Discussion manuscript review of: "Radiative effects of ozone waves on the Northern Hemisphere polar vortex and its modulation by the QBO" By: V. Silverman et al.

Let me begin by saying that I very much like the paper. The approach using wave packets and taking into the account the implications of the seasonal cycle are novel and lead to insightful results. To be honest, I wish that I had more critical and helpful things to say, but for the most part, the conclusions are physically based and sound. Generally speaking, the paper is well-written, but there are some grammatical issues that need fixing (I don't think I commented on all of the grammar/spelling issues, so

please go over the paper carefully and correct any additional misspellings and errors that I missed). If the authors can take into account my relatively short list of minor suggestions below, then I will gladly recommend this paper for publication.

Major comments:

Comment #1 – Page 3 lines 22-23: I'm not sure about the seasonality statement here. You should double check, but if I recall correctly, Watson and Gray (JAS 2014) find that the QBO signal is stronger later in the winter. This may be an important point in light of the fact that your argument hinges on the seasonal cycle of the waves and the mean. If I am correct here, it would be good for you to comment on how Watson and Gray's results apply to your study.

Comment #2 – Page 4 lines 15-20: How does your approach deal with ozone flux convergences in the ZMO3 runs? While I understand that you only pass zonally symmetrized ozone to the radiation code, the zonal mean ozone does still include one effect of ozone waves on the simulations if the zonal mean ozone field includes the flux convergences. You should clarify this one way or the other and make it clear to readers exactly what pieces of wave ozone physics are included in each type of simulation (i.e. 3DO3 versus ZMO3).

Comment #3 – Page 30 line 30: You mention later that your results are robust to the 70th percentile choice, but I am wondering about the 100 hPa level. I say this because the 100 hPa level is a very sensitive region in the stratosphere as far as the "valving" of wave energy either upwards into the core of the vortex where the PV gradient is strong and there is a strong waveguide versus ducting the energy equatorward. I am guessing that your results are robust to this choice, but it would be good for readers to know this information. I say this mostly because I think your approach is novel and it would be good for readers to be able to have all of the information they need to apply the method in other contexts.

Comment #4 – Page 5 lines 14: Sorry to be picky, but I really think that you should include the original source here when discussing the inverse relationship between ozone and temperature, which is Craig and Ohring 1958, see citation below:

http://journals.ametsoc.org/doi/abs/ 10.1175/1520-0469%281958%29015%3C0059%3ATTDOOR%3E2.0.CO%3B2

Also, while the Hartmann 1981 paper is nice in a qualitative sense, much more detailed information can be gathered from the following sets of papers that I think you should also cite: Nathan and Cordero JGR 2007, Hartmann and Garcia JAS 1979, and Garcia and Hartmann JAS 1980. I think in particular the Garcia references are important because they are directly relevant to the physical interpretations of your work and have a good amount of physical insight in them that readers should know about.

Comment #5 – Page 5 lines 10-30:Two related issues here. One, there is some seasonality to the ratio of advective to photochemical timescales and the ratio of advective to Newtonian cooling timescales (see Fig. 3 of Nathan and Cordero JGR 2007). Also, there is strong seasonality in regards to many wave properties as outlined carefully in Nathan and Li (JAS 1991) and Nathan and Cordero (JGR 2007). Do your results agree with these theoretical results? While this may not be a simple set of questions to answer, I think that lending some effort towards deciphering if your WACCM results agree with previous theory would be nice. I will leave it up to you on where you want to comment on this (perhaps the results section is not the right place), but it would be helpful if you could comment somewhere in your text.

Comment #6 – Page 8 lines 25-30: Why are you using the beta-plane geometry form instead of the spherical form? I am wondering if your figure would look any different using the full form. I am also wondering a bit about your interpretation of the refractive index (RI) anomalies. In particular, while I do find your point regarding the ducting of wave energy in the middle portion of the domain (i.e. the blue region spanning 15-45 km in height and 70-80 N to 20 N) during west QBO, I am wondering about your interpretation during east QBO. That is, while there is a region of positive RI in the uppermost stratosphere during east QBO, before the wave energy gets there, it would

first encounter the broad region of negative RI anomaly (i.e. the same blue region I just described above). And given that there appears to be a region of positive RI immediately underneath the blue region (i.e. the red region extending from 60 N to 30 N between 10-30 km in height), isn't it possible that a bunch of wave energy is also being ducted equatorward during east QBO (but lower than is being ducted during QBO west)? Indeed it is somewhat hard to tell from Fig. 8c, but it seems like there is additional EP-flux convergence near 30-40 N at 30 km for QBO east. I'm not saying that there is any inconsistency in your argument, but perhaps east QBO is characterized by both increased upper stratospheric convergence and subtropical convergence at 30 km. Just a thought. Would the spherical form of the RI make determining this clearer? What about the individual wavenumber diagnostics (see below)?

Also, just out of curiosity, why are you not diagnosing the individual wavenumbers as per Eqs. (12) and (13) in Harnik and Lindzen (2001)? I'm certainly okay with using the more traditional 'Matsuno-like' RI and so I am not demanding that you use the individual wavenumber method, rather I am actually just curious for the rationale.

Comment #7 – Page 9 lines 19-20: Why exactly is it expected that the nonlinear terms are larger during QBO east? I realize that the QBO east is characterized by more wave driving, but couldn't that appear via the quasi-nonlinear PV flux term (1st term on the RHS of eq. 1) and not via the fully nonlinear terms? I realize that you cite the White et al. (2016) paper in the next sentence, but that just means that your results are consistent. Stating that something is "as expected" seems to imply that there is a physical reason to expect this result.

Comment #8 – Page 9 lines 25-28: If I understand your line of reasoning here, you are stating the ZMO3 run has stronger damping in the lower stratosphere and weaker damping in the upper stratosphere. Or said another way, 3d ozone decreases ozone damping in the lower stratosphere but increases damping in the upper stratosphere. You mention in Section 3.1 some of the ozone physics involved, but then you don't mention any of that here. I would say that something interesting can be said regarding

what is happening. My initial take would be the following (though for sure the authors should give their own interpretation of the results because I may be missing something).

(Note that the discussion below also has implications for your results on page 10 lines 29-35 through page 11 lines 1-9).

Based on photochemical and dynamical timescales, the 3d ozone induced decrease in damping in the lower stratosphere must be associated with advection of zonal mean ozone by the wave fields, yes? And in the upper stratosphere, the 3d ozone induced increase in damping is due to photochemistry, yes? Now, the upper stratospheric increase in damping is to be expected based on the ozone-temperature phase relationship dictated by the temperature dependent Chapman chemistry (e.g., Craig and Ohring 1958).

However, the lower stratospheric dynamically-based ozone result is fundamentally dependent on the vertical and horizontal ozone gradients. Previous studies have discussed this bit of physics but only in the context of 1D mechanistic models (e.g., Nathan and Cordero 2007 and Albers and Nathan 2012). However, your results are the first to be able to state something more general and thus it may be worth pointing out that it appears that 3d ozone causes dynamically induced ozone heating anomalies that decrease wave damping. This would mean that if there is any seasonal cycle to the vertical and meridional ozone gradients, then there should be some seasonality to the effect of 3d ozone that is perhaps contributing to the enhancement of the HT effect that you describe in your conclusions. Or perhaps the vertical and meridional ozone gradients are different for the wQBO versus eQBO, which in turn leads to some of the differences you see in the EP-flux divergence for the two QBO phases? To be honest, I don't have this all worked out in my head clearly, but it is perhaps worth thinking about because it would seem you might be able to add some physical insight here in the context of a CCM whereas previous studies with physics discussions where limited because of their model simplicity. I should also mention that you can quite easily see

how all of the ozone physics modulate the EP-flux divergence by considering Eq. (14) in combination with Eq. (15) (for the lower stratosphere) and Eq. (17) (for the upper stratosphere) in Nathan and Cordero (2007).

Comment #9 – Page 9 Equation (1): Please define your notation here and don't just cite Smith (1983). Specifically, do the different primes mean something different? That is, do the primes in the PV flux term (1st term on the RHS) somehow denote something different from than the primes in the nonlinear terms (2nd and 3rd terms on the RHS)?

Minor comments:

Comment #1 – Introduction lines 1-2: "...exist since the early..." should be "...have existed since the early..."

Comment #2 – Page 2 line 1: Multi decadal should be hyphenated as multi-decadal.

Comment #3 – Page 2 line 4: "...(Taylor et al. 2012), does not..." should be "...... (Taylor et al. 2012), do not..."

Comment #4 – Page 2 line 2: While I could be wrong, I believe that you meant to use the word "assess" and not the word "asses" :)

Comment #5 – Page 3 line 3: Using a hyphen here doesn't work grammatically. Please rework this sentence.

Comment #6 – Page 3 line 17: I would suggest also citing the new (ish) paper by Watson and Gray (JAS January 2014) because it provides new insights supporting the original HT-1980 paper, which Garfinkel et al. 2012 (which you cite) call into question.

Comment #7 – Page 3 line 21: "noninear" should be "nonlinear".

Comment #8 – Page 4 line 24: "tendenfcy" should be "tendency"

Comment #9 – Page 3 lines 24-25: Which figures are you referring to? This is a bit vague.

Comment #10 – Page 5 line 15: Capitalize "northern hemisphere" (both words)

Comment #11 – Page 5 line 21: Similar to my Major Comment #4, while the Douglass reference is nice, I really think that the Hartmann/Garcia 1979 and Garcia/Hartmann 1980 references are very relevant here and they pre-date the Douglass reference by half a decade. They should also be included.

Comment #12 – Page 9 lines 3-4: You seem to be stating the same thing twice here (regarding non-acceleration conditions).

---

## Referee Comment (RC2) · Anonymous Referee #2 · 11 Sep 2017

9/11/2017

Radiative effects of ozone waves on the Northern Hemisphere polar vortex and its modulation by the QBO by Silverman et al Recommendation: minor revisions

This paper discusses the interaction of ozone waves with the dynamics of the fall and winter Northern Hemisphere polar stratosphere in WACCM. Consistent with previous work, the sign of the ozone-temperature wave correlation differs between the lower and upper stratosphere. This leads to differences in wave damping and in the fall vortex. If EQBO and WQBO winters are composited separately, then differences are evident in mid-winter as well due to the differing effects of each QBO phase on wave propagation.

[Figure]

The results are generally sound and convincing. The heat flux pulse methodology is novel and might applicable in other contexts as well. I have one potentially substantive comment about their key figure, and assuming that the relationship in this key figure is statistically significant all other changes are minor.

General Comments:

1. Much of this paper is based on differences between the top half and bottom half of figure 4. However, the authors don't appear to have explicitly calculated the statistical significance of the difference between them. The authors need to confirm that the difference between panels 4a and panels 4c, and likewise between panels 4b and 4d, is actually statistically significant.

2. I found figure 12 and its accompanying paragraph to be very confusing. What exactly is Fyy? The y-derivative of the y-component of EP flux? Similar what is Fzz? The z-derivative of the z-component of EP flux? Even if I assume this to be the case, I had serious trouble following the text and the accompanying figure despite multiple rereads. Either the authors need to expand their discussion and help the reader a bit, or remove this entirely as it doesn't appear to be crucial for the rest of the paper.

Technical comments: The abstract was quite long and wordy. It can almost certainly be shortened without removing key content.

P1, line 8 "in the natural configuration" can be removed. While this may have meaning to someone within the NCAR world, it has little meaning to someone on the outside

P2 line 1 chemistry climate models are also used in air pollution studies and for aerosol studies. See the AER-CHEM-MIP project (https://wiki.met.no/aerocom/aerchemmip/start)

P2 line 4 the majority . . . . do not

P2 line 29 this paragraph extends for 31 lines and is hard to digest! I suggest adding two new paragraph breaks: a first on line 4 of page 3, before "Also", and a second on

line 24 of page 3 before "To understand"

P3 line 21 nonlinear is misspelled

P4 line 24 tendency is misspelled

P4 line 26 I do not understand this sentence. Please rewrite

P4 line 28 ozone wave**s**

It may be helpful to add an intro sentence to section 3, rather than diving straight into the nitty gritty of the results

P8 line 22 composites is misspelled

P9 line 4 sentence is repeated

P10 line 34 "descends lower down" it is impossible to infer this from figure 13. This clause should either be removed, or reference made to a different figure.

P 11 line 2 the second half of this sentence is very unclear and needs to be rewritten

P 11 line 27 I suggest starting a new paragraph with "While"

Figure 1 units are not indicated on the colorbar on the left column

Figure 5 is missing units

Figure 8 is missing the x-label (latitude)

Figure 9 either the caption or the figure itself should state explicitly EQBO-WQBO

Figure 13: The caption should note that a thick line indicates statistical significance (assuming I infer correctly).
* * *

---

## Author Comment (AC1) · 20 Nov 2017

**Radiative effects of ozone waves on the Northern Hemisphere polar vortex and its modulation by the QBO**

Vered Silverman[1], Nili Harnik[1], Katja Matthes[2], Sandro W. Lubis[3], and Sebastian Wahl[2]

[1]Department of Geophysical, Atmospheric and Planetary Sciences, Tel Aviv University, Tel Aviv, Israel
[2]GEOMAR Helmholtz Centre for Ocean Research Kiel, Kiel, Germany
[3]Department of the Geophysical Sciences, The University of Chicago, USA

*Correspondence to:* Vered Silverman (veredsil@post.tau.ac.il)

**Reviewer #1**

We thank the reviewer for reading the manuscript and providing their helpful comments. We address their issues below.

– Comment #1 – Page 3 lines 22-23: I'm not sure about the seasonality statement here. You should double check, but if I recall correctly, Watson and Gray (JAS 2014) find that the QBO signal is stronger later in the winter. This may be an important point in light of the fact that your argument hinges on the seasonal cycle of the waves and the mean. If I am correct here, it would be good for you to comment on how Watson and Gray's results apply to your study.

– Answer #1 - Watson and Gray (2014) indeed find a later Holton-Tan signal in their model, compared to our study and to ERA40, where it appears earlier in November. We note that Watson and Gray (2014) use the HadGEM2-CCS model, which is not coupled to chemistry, and it seems that zonally averaged ozone is used in the radiative scheme. Therefore, the lack of an early winter QBO response in their model is in actually in agreement with our results. We added this in the conclusions section. Thank you.

– Comment #2 – Page 4 lines 15-20: How does your approach deal with ozone flux convergences in the ZMO3 runs? While I understand that you only pass zonally symmetric ozone to the radiation code, the zonal mean ozone does still include one effect of ozone waves on the simulations if the zonal mean ozone field includes the flux convergences. You should clarify this one way or the other and make it clear to readers exactly what pieces of wave ozone physics are included in each type of simulation (i.e. 3DO3 versus ZMO3).

– Answer #2- We intentionally keep the influence of ozone waves on the zonal mean ozone through advection, as we are interested only in the direct radiative effect of ozone which is the influence on Newtonian damping. We further clarified this on Page 4 line 25-26.

– Comment #3 – Page 30 line 30: You mention later that your results are robust to the 70th percentile choice, but I am wondering about the 100 hPa level. I say this because the 100 hPa level is a very sensitive region in the stratosphere as

far as the "valving" of wave energy either upwards into the core of the vortex where the PV gradient is strong and there is a strong waveguide versus ducting the energy equatorward. I am guessing that your results are robust to this choice, but it would be good for readers to know this information. I say this mostly because I think your approach is novel and it would be good for readers to be able to have all of the information they need to apply the method in other contexts.

5    – Answer #3- We repeated the analysis for events chosen using the 50hPa level and the results are qualitatively similar (see Figures 1,2 below). We chose the 100hPa level since this is the region where the waves enter the stratosphere, and we wanted to identify the events before the wave changes the mean flow. This is mentioned briefly in the text on page 5 line 11-12.

   – Comment #4 – Page 5 lines 14: Sorry to be picky, but I really think that you should include the original source here
10       when discussing the inverse relationship between ozone and temperature, which is Craig and Ohring 1958, see citation below: http://journals.ametsoc.org/doi/abs/10.1175/1520-0469%281958%29015%3C0059%3ATTDOOR%3E2.0.CO% 3B2 Also, while the Hartmann 1981 paper is nice in a qualitative sense, much more detailed information can be gathered from the following sets of papers that I think you should also cite: Nathan and Cordero JGR 2007, Hartmann and Garcia JAS 1979, and Garcia and Hartmann JAS 1980. I think in particular the Garcia references are important because they
15       are directly relevant to the physical interpretations of your work and have a good amount of physical insight in them that readers should know about.

   – Answer #4 - We replaced the reference to Craig and Ohring (1958), and thank the reviewer for this correction. We also added a sentence on why the ozone-temperature correlation is positive in the dynamically controlled region, with a reference to Hartmann and Garcia (1979) (page 6 line 5-6) and to Nathan and Cordero (2007) (page 6 line 18-19, page
20       13 line 21-25). Since this is not the topic of the paper, we decided not to elaborate any further.

   – Comment #5 – Page 5 lines 10-30:Two related issues here. One, there is some seasonality to the ratio of advective to photochemical timescales and the ratio of advective to Newtonian cooling timescales (see Fig. 3 of Nathan and Cordero JGR 2007). Also, there is strong seasonality in regards to many wave properties as outlined carefully in Nathan and Li (JAS 1991) and Nathan and Cordero (JGR 2007). Do your results agree with these theoretical results? While this may
25       not be a simple set of questions to answer, I think that lending some effort towards deciphering if your WACCM results agree with previous theory would be nice. I will leave it up to you on where you want to comment on this (perhaps the results section is not the right place), but it would be helpful if you could comment somewhere in your text.

   – Answer #5 - Nathan and Li (1991) showed that ozone wave effects are strongest in September, and weakest during January due to the large solar zenith angle. Also, when the waves peak in the lower stratosphere, where the ozone-
30       temperature correlation is positive, the dominant ozone wave effect is to weaken the radiative damping in that region. This is indeed seen in September, where the temperature wave is stronger throughout the stratosphere in the 3DO3 run (we added a comment on this on page 6 line 30-32). On the other hand, when the waves penetrate higher into the upper stratosphere, where the ozone-temperature correlation is negative, the dominant wave radiative effect is to strengthen

the thermal damping. As we show in Figure 3 - during September the wave peak is in the region where the ozone-temperature correlation is positive, thus the main effects are weaker wave damping, stronger waves (showed by the positive |T| anomalies), and correspondingly, a vertical displacement of the EP flux peak. In the WACCM model, it does not seem like this changes later in winter. Apart from the obvious model differences (1D vs CCM) it is possible this is also due to the limited height range (up to 2hPa) over which we zonally average the ozone field in the ZMO3 run in order to avoid large biases in the mesosphere. We added a comment to this effect on page 8 line 2-8.

– Comment #6 – Page 8 lines 25-30: Why are you using the beta-plane geometry form instead of the spherical form? I am wondering if your figure would look any different using the full form. I am also wondering a bit about your interpretation of the refractive index (RI) anomalies. In particular, while I do find your point regarding the ducting of wave energy in the middle portion of the domain (i.e. the blue region spanning 15-45 km in height and 70-80 N to 20 N) during west QBO, I am wondering about your interpretation during east QBO. That is, while there is a region of positive RI in the uppermost stratosphere during east QBO, before the wave energy gets there, it would first encounter the broad region of negative RI anomaly (i.e. the same blue region I just described above). And given that there appears to be a region of positive RI immediately underneath the blue region (i.e. the red region extending from 60 N to 30 N between 10-30 km in height), isn't it possible that a bunch of wave energy is also being ducted equatorward during east QBO (but lower than is being ducted during QBO west)? Indeed it is somewhat hard to tell from Fig. 8c, but it seems like there is additional EP-flux convergence near 30-40 N at 30 km for QBO east. I'm not saying that there is any inconsistency in your argument, but perhaps east QBO is characterized by both increased upper stratospheric convergence and subtropical convergence at 30km. Just a thought. Would the spherical form of the RI make determining this clearer? What about the individual wavenumber diagnostics (see below)? Also, just out of curiosity, why are you not diagnosing the individual wavenumbers as per Eqs. (12) and (13) in Harnik and Lindzen (2001)? I'm certainly okay with using the more traditional 'Matsuno-like' RI and so I am not demanding that you use the individual wavenumber method, rather I am actually just curious for the rationale.

– Answer #6 - We actually used the spherical form of the index of refraction, and had a typo in the caption of Figure 9, which we corrected. Figure 3 below shows the index of refraction (top), meridional (middle) and vertical (bottom) wavenumbers for east, west, and east minus west QBO phases. The differences in the index of refraction are dominated by the differences in vertical wavenumber. The reviewer is right that there is a stronger meridional wavenumber anomaly in the subtropics (30-40N, 20-30km) during east QBO, however this does not have a very strong signal in the EP flux field on days -10 to -5 (see paper Fig 8). The vertical wavenumber clearly dominates the IR anomalies even in the subtropics, and as indicated by other measures, the increased vertical propagation in east QBO is the main difference. We added a comment on this on page 9 line 24-25.

– Comment #7 –Page 9 lines 19-20: Why exactly is it expected that the nonlinear terms are larger during QBO east? I realize that the QBO east is characterized by more wave driving, but couldn't that appear via the quasi-nonlinear PV flux term (1st term on the RHS of eq. 1) and not via the fully nonlinear terms? I realize that you cite the White et al. (2016)

paper in the next sentence, but that just means that your results are consistent. Stating that something is "as expected" seems to imply that there is a physical reason to expect this result.

– Answer #7 - We expected the the nonlinear terms to be larger in the east QBO as we see these events are less reversible. We changed the text to make this point clearer. (page 9 line 17 to page 10 line 2).

– Comment #8 – Page 9 lines 25-28: If I understand your line of reasoning here, you are stating the ZMO3 run has stronger damping in the lower stratosphere and weaker damping in the upper stratosphere. Or said another way, 3d ozone decreases ozone damping in the lower stratosphere but increases damping in the upper stratosphere. You mention in Section 3.1 some of the ozone physics involved, but then you don't mention any of that here. I would say that something interesting can be said regarding what is happening. My initial take would be the following (though for sure the authors should give their own interpretation of the results because I may be missing something). (Note that the discussion below also has implications for your results on page 10 lines 29-35 through page 11 lines 1-9). Based on photochemical and dynamical timescales, the 3d ozone induced decrease in damping in the lower stratosphere must be associated with advection of zonal mean ozone by the wave fields, yes? And in the upper stratosphere, the 3d ozone induced increase in damping is due to photochemistry, yes? Now, the upper stratospheric increase in damping is to be expected based on the ozone-temperature phase relationship dictated by the temperature dependent Chapman chemistry (e.g., Craig and Ohring 1958). However, the lower stratospheric dynamically-based ozone result is fundamentally dependent on the vertical and horizontal ozone gradients. Previous studies have discussed this bit of physics but only in the context of 1D mechanistic models (e.g., Nathan and Cordero 2007 and Albers and Nathan 2012). However, your results are the first to be able to state something more general and thus it may be worth pointing out that it appears that 3d ozone causes dynamically induced ozone heating anomalies that decrease wave damping. This would mean that if there is any seasonal cycle to the vertical and meridional ozone gradients, then there should be some seasonality to the effect of 3d ozone that is perhaps contributing to the enhancement of the HT effect that you describe in your conclusions. Or perhaps the vertical and meridional ozone gradients are different for the wQBO versus eQBO, which in turn leads to some of the differences you see in the EP-flux divergence for the two QBO phases? To be honest, I don't have this all worked out in my head clearly, but it is perhaps worth thinking about because it would seem you might be able to add some physical insight here in the context of a CCM whereas previous studies with physics discussions where limited because of their model simplicity. I should also mention that you can quite easily see how all of the ozone physics modulate the EP-flux divergence by considering Eq. (14) in combination with Eq. (15) (for the lower stratosphere) and Eq. (17) (for the upper stratosphere) in Nathan and Cordero (2007).

– Answer #8 - The equations for ozone-modified refractive index of Nathan and Cordero (2007) are very insightful for the simplified model, however, in our case, the analysis is complicated by a rich latitude-height structure, and we did not gain a simplified understanding of the role of specific terms.

We distinguish in our answer between a few effects.

1. Zonal mean ozone feedbacks: the effects we discuss in the paper, which amplify the initial radiative perturbation, via a modulation of wave propagation, to a change in the polar vortex and the wave induced overturning circulation, will also change the zonal mean ozone gradients. These changes will feed back onto the initial radiative perturbation we imposed. The sign of this effect, however, is unclear, for the following reason. The ozone induced radiative heating is proportional to the ozone wave amplitude and to the correlation between ozone and temperature waves. The ozone wave 1 amplitude time tendency is dominated by meridional advection (the peaks of ozone wave 1 amplitude follow the peaks in the meridional gradient of the zonal mean ozone (Fig. 4), and the meridional advection term ($v'\frac{d\overline{O3}}{dY}$) is the strongest term in the budget. In the 3DO3 run the meridional gradient is weaker compared to the ZMO3 run. This changes the ozone wave amplitude but not in a straightforward way, because the correlation between ozone and meridional wind also changes, so that the change in ozone wave 1 amplitude has a complex structure with the most significant feature being a weakening at the upper part of the ozone wave peak. This weakening is at the level at which the ozone waves transition from dynamical to radiative control and thus the effect on radiative wave damping is small. At the same time, the lower stratospheric correlation between ozone and temperature wave fields is slightly more positive in the 3DO3 run (Fig. 5), which will result in a weakening of the thermal damping. Thus the sign of the feedback is not clear.

2. A seasonality in the zonal mean ozone and ozone-temperature wave correlation fields: Examining these fields, we find the changes small, consistent with the dominant term in the seasonality of the direct radiative effect being due to the change in zenith angle.

3. A difference in the zonal mean ozone field and ozone-temperature wave correlations between east and west QBO: We repeat the analysis of section 3.2, compositing different fields centered around October upward wave pulses, for east and west QBO phases separately. Fig. 6a shows the life-cycle mean (days -10 to 15) of zonal mean ozone gradients. We see that the largest differences are in the tropical and subtropical region, where the fields are directly forced by the QBO. This causes a weaker ozone wave 1 amplitude in that region during EQBO (Fig 6b). The weaker meridional gradient in the high latitude region also causes a smaller wave amplitude of ozone wave 1 during EQBO (Fig. 6b), resulting in a weaker ozone direct radiative effect in the lower stratosphere - stronger damping during EQBO events (Fig. 6c). This is also accompanied by a slightly weaker ozone-temperature correlation (Fig. 6d). The temperature wave amplitude, however, is still stronger for east QBO events in the mid-upper stratosphere (Fig. 6e). This suggests the QBO induced changes in the zonal mean ozone field are of secondary order.

These results were partially added on page 6 line 8-9, page 11 lines 16-23.

– Comment #9 – Page 9 Equation (1): Please define your notation here and don't just cite Smith (1983). Specifically, do the different primes mean something different? That is, do the primes in the PV flux term (1st term on the RHS) somehow denote something different from than the primes in the nonlinear terms (2nd and 3rd terms on the RHS)?

– Answer #9 - We took care to define everything carefully and fixed the use of different primes- they were meant to be the same. Besides this we did not find any terms which are not defined.

Minor Comments

– Comment #1 – Introduction lines 1-2: "...exist since the early..." should be "...have existed since the early..."

– Answer #1 - fixed

– Comment #2 – Page 2 line 1: Multi decadal should be hyphenated as multi-decadal.

– Answer #2 - fixed

– Comment #3 – Page 2 line 4: "...(Taylor et al. 2012), does not..." should be "...... (Taylor et al. 2012), do not..."

– Answer #3 - fixed

– Comment #4 – Page 2 line 2: While I could be wrong, I believe that you meant to use the word "assess" and not the word "asses" :)

– Answer #4 - fixed

– Comment #5 – Page 3 line 3: Using a hyphen here doesn't work grammatically. Please rework this sentence.

– Answer #5 - fixed

– Comment #6 – Page 3 line 17: I would suggest also citing the new (ish) paper by Watson and Gray (JAS January 2014) because it provides new insights supporting the original HT-1980 paper, which Garfinkel et al. 2012 (which you cite) call into question.

– Answer #6 - added

– Comment #7 – Page 3 line 21: "noninear" should be "nonlinear".

– Answer #7 - fixed

– Comment #8 – Page 4 line 24: "tendenfcy" should be "tendency"

– Answer #8 - fixed

– Comment #9 – Page 3 lines 24-25: Which figures are you referring to? This is a bit vague.

– Answer #9 - fixed

– Comment #10 – Page 5 line 15: Capitalize "northern hemisphere" (both words)

- Answer #10 - fixed

- Comment #11 – Page 5 line 21: Similar to my Major Comment #4, while the Douglass reference is nice, I really think that the HartmannGarcia 1979 and GarciaHartmann 1980 references are very relevant here and they pre-date the Douglass reference by half a decade. They should also be included.

- Answer #11 -

- Comment #12 – Page 9 lines 3-4: You seem to be stating the same thing twice here (regarding non-acceleration conditions).

- Answer #12 - fixed

[Figure]

**Figure 1.** Lat-height time lag composites of EP-flux divergence anomalies from the climatology) for the positive heat flux EQBO (top), WQBO (mid), and the difference between them (bot), for October events (70th percentile of $\overline{V'T'}$ at 50mb 85-45N) of the 3DO3 run. Statistically significant areas are shown by gray shading.

[Figure]

**Figure 2.** Lat-height time lag composites of EP-flux divergence anomalies from the climatology) for the positive heat flux EQBO (top), WQBO (mid), and the difference between them (bot), for October events (70th percentile of $\overline{V'T'}$ at 50mb 85-45N) of the ZMO3 run. Statistically significant areas are shown by gray shading.

[Figure]

**Figure 3.** (Top) Index of refraction $\left(n^2 = N^2 \left[ \frac{a\overline{q}_y}{\overline{U}-c} - \frac{s^2}{cos^2\phi} + a^2 f^2 F(N^2) \right]\right)$, see eq.C2,5 in Harnik and Lindzen (2001))at days $-10$ to $-5$ for east (left), west (center) and the difference between east and west QBO (right) in the 3DO3 run. The meridional and vertical wave components are shown in the middle and bottom row, correspondingly.

[Figure]

**Figure 4.** Meridional gradient of the zonal mean ozone (color), ozone wave 1 amplitude (green contours) and temperature wave 1 amplitude (gray contours), for Sep-Nov.

[Figure]

**Figure 5.** Latitude-height of 3DO3-ZMO3 (colors) of (a) Meridional gradient of the zonal mean ozone, (b) wave 1 amplitude of the ozone tendency from advection, (c) ozone-temperature correlation for zonal wave 1, (d) ozone wave 1 amplitude and (e) ozone-meridional wind correlation for zonal wave 1, for Sep-Nov. Climatology of the 3DO3 run is shown in green contours.

[Figure]

(a) $\frac{1}{a}\frac{d\overline{O_3}}{d\phi}$

(b) $O_3$ wv1 amplitude

(c) Temperature tendency from SWR, normalized by $|T_{wv1}|$

(d) $cos(\theta_T - \theta_{O_3})$

(e) Temperature wv1 amplitude

**Figure 6.** Latitude-height of E-W QBO October positive hat flux events composit for day -10 to 15 (a) Meridional gradient of the zonal mean ozone, (b) wave 1 amplitude of the ozone, (c) temperature wave 1 amplitude tendency from short-wave radiation, (d) ozone-temperature correlation for zonal wave 1, and (e) temperature wave 1 amplitude.

**References**

Craig, R. A. and Ohring, G.: The temperature dependence of ozone radiational heating rates in the vicinity of the mesopeak, Journal of Meteorology, 15, 59–62, 1958.

Harnik, N. and Lindzen, R. S.: The effect of reflecting surfaces on the vertical structure and variability of stratospheric planetary waves, Journal of the atmospheric sciences, 58, 2872–2894, 2001.

Hartmann, D. and Garcia, R.: A Mechanistic Model of Ozone Transport by Planetary Waves in the Stratosphere, J. Atmos. Sci., 36, 350–364, 1979.

Nathan, T. R. and Cordero, E. C.: An ozone-modified refractive index for vertically propagating planetary waves, Journal of Geophysical Research: Atmospheres, 112, 2007.

Nathan, T. R. and Li, L.: Linear stability of free planetary waves in the presence of radiative–photochemical feedbacks, Journal of the atmospheric sciences, 48, 1837–1855, 1991.

Watson, P. A. and Gray, L. J.: How does the quasi-biennial oscillation affect the stratospheric polar vortex?, Journal of the Atmospheric Sciences, 71, 391–409, 2014.

---

## Author Comment (AC2) · 20 Nov 2017

**Radiative effects of ozone waves on the Northern Hemisphere polar vortex and its modulation by the QBO**

Vered Silverman[1], Nili Harnik[1], Katja Matthes[2], Sandro W. Lubis[3], and Sebastian Wahl[2]

[1]Department of Geophysical, Atmospheric and Planetary Sciences, Tel Aviv University, Tel Aviv, Israel
[2]GEOMAR Helmholtz Centre for Ocean Research Kiel, Kiel, Germany
[3]Department of the Geophysical Sciences, The University of Chicago, USA

*Correspondence to:* Vered Silverman (veredsil@post.tau.ac.il)

**Reviewer #2**

We thank the reviewer for reading the manuscript and providing their helpful comments. We address their issues below.

– Comment #1 - Much of this paper is based on differences between the top half and bottom half of figure 4. However, the authors don't appear to have explicitly calculated the statistical significance of the difference between them. The authors need to confirm that the difference between panels 4a and panels 4c, and likewise between panels 4b and 4d, is actually statistically significant.

– Answer #1- To perform this significance test it would be ideal to have many realizations of each simulation (3D or ZM ozone in the radiation code). Since this is not possible with our resources, we do the following:

1. Take all years of the two simulations - a total of 200 years, and mix them together to one data set of 200 winters.

2. Randomly choose four groups of winters according to the number of east/west QBO winters for each run (two groups for 3DO3 and two for the ZMO3 run).

3. Average each group and take the difference between the east and west groups for each simulation.

4. Repeat this 1000 times to get a statistical distribution of 3DO3 minus ZMO3 east minus west QBO anomalies, for each latitude and height grid point.

We now have 1000 differences of random winters for each run. Statistical significance of the 3D(E-W) and ZM(E-W) is calculated by checking if the difference of the E-W (3D-ZM) is bigger/smaller than the 97.5/2.5 percentile of the difference between the two distributions we got.

The result of this calculation is shown in Figure 1 for the zonal mean zonal wind (top) and zonal mean temperature (bot). In the zonal mean zonal winds the negative/positive values in early/late winter indicate that the E-W difference in the 3DO3 run is stronger/weaker than the E-W difference in the ZMO3 run, corresponding to a delay in the HT signal. The

differences are statistically significant. The delayed HT signal in the zonal mean temperature is statistically significant as well.

This information was added in the Appendix section.

- Comment #2 - I found figure 12 and its accompanying paragraph to be very confusing. What exactly is Fyy? The y-derivative of the y-component of EP flux? Similar what is Fzz? The z-derivative of the z-component of EP flux? Even if I assume this to be the case, I had serious trouble following the text and the accompanying figure despite multiple rereads. Either the authors need to expand their discussion and help the reader a bit, or remove this entirely as it doesn't appear to be crucial for the rest of the paper.

- Answer #2 - Fyy/Fzz are indeed the y-derivative / z-derivative of the y-component /z-component of EP flux. Following this comment we decided to remove this figure and replaced it with a new one (Figure 12) showing only the EP flux divergence differences between the 3DO3 and ZMO3 runs for east/west QBO events. The relevant text is updated on page 11 line 10-16.

Minor comments:

- Technical comments: The abstract was quite long and wordy. It can almost certainly be shortened without removing key content.

- Answer - the abstract has been re-written.

- Comment #1 - P1, line 8 "in the natural configuration" can be removed. While this may have meaning to someone within the NCAR world, it has little meaning to someone on the outside

- Answer #1 - fixed

- Comment #2 - P2 line 1 chemistry climate models are also used in air pollution studies and for aerosol studies. See the AER-CHEM-MIP project (https://wiki.met.no/aerocom/aerchemmip/start)

- Answer #2 - fixed

- Comment #3 - P2 line 4 the majority . . .. do not

- Answer #3 - fixed

- Comment #4 - P2 line 29 this paragraph extends for 31 lines and is hard to digest! I suggest adding two new paragraph breaks: a first on line 4 of page 3, before "Also", and a second on line 24 of page 3 before "To understand"

- Answer #4 - fixed

- Comment #5 - P3 line 21 nonlinear is misspelled

- Answer #5 - fixed

- Comment #6 - P4 line 24 tendency is misspelled

- Answer #6 - fixed

- Comment #7 - P4 line 26 I do not understand this sentence. Please rewrite

- Answer #7 - fixed

- Comment #8 - P4 line 28 ozone wave**s**

- Answer #8 - fixed

- Comment #9 - It may be helpful to add an intro sentence to section 3, rather than diving straight into the nitty gritty of the results

- Answer #9 - added

- Comment #10 - P8 line 22 composites is misspelled

- Answer #10 - fixed

- Comment #11 - P9 line 4 sentence is repeated

- Answer #11 - fixed

- Comment #12 - P10 line 34 "descends lower down" it is impossible to infer this from figure 13. This clause should either be removed, or reference made to a different figure.

- Answer #12 - fixed

- Comment #13 - P 11 line 2 the second half of this sentence is very unclear and needs to be rewritten

- Answer #13 - fixed

- Comment #14 - P 11 line 27 I suggest starting a new paragraph with "While"

- Answer #14 - fixed

- Comment #15 - Figure 1 units are not indicated on the colorbar on the left column

- Answer #15 - fixed

- Comment #16 - Figure 5 is missing units

- Answer #16 - fixed

(a) $U_{zm}$, (E-W)QBO 3DO3-ZMO3

(b) $T_{zm}$, (E-W)QBO 3DO3-ZMO33

**Figure 1.** Daily climatology east-west QBO differences between the 3DO3 and ZMO3 model runs of the zonal mean zonal wind averaged over 75-55N (top) and the zonal mean temperature averaged over 90-66N for the 3DO3 (bot), for Sep-Mar. Statistically significant areas are shown by gray shading.

- – Comment #17 - Figure 8 is missing the x-label (latitude)

- – Answer #17 - fixed (added to last row of the figure, is it enough?)

- – Comment #18 - Figure 9 either the caption or the figure itself should state explicitly EQBO-WQBO

- – Answer #18 - added to text

- – Comment #19 - Figure 13: The caption should note that a thick line indicates statistical significance (assuming I infer correctly).

- – Answer #19 - fixed

**References**

---

## Referee Report (RR1)

***Atmospheric and Chemistry and Physics Discussion manuscript review of***:
*"Radiative effects of ozone waves on the Northern Hemisphere polar vortex and its modulation by the QBO"*
*By: V. Silverman et al.*

The authors have (for the most part) addressed my questions and concerns, with one exception (detailed below). If the authors address this issue, then I think the manuscript is ready for publication.

**Minor comment (referring to line #'s 20-30 on page 5 of the version 3 manuscript) :**

In my previous review, I asked if the authors could clarify exactly what types of ozone physics are retained and suppressed in the 3DO3 versus ZMO3 runs. The authors tried to address this but it is still a little unclear to me what they did. On the one hand, the authors added a reference to the figure in the Albers/Nathan paper, which helps clarify that they are primarily interested in understanding the effects of photochemical wave damping via the inclusion of three-dimensional ozone in the radiation code calculations. However, the authors still do not clarify whether their ZMO3 runs include what Albers/Nathan refer to as "pathway two". That is, in the ZMO3 runs, does the advection scheme *only* advect ozone via zonal mean processes?  In other words, if the advection scheme is three-dimensional, then the zonal mean ozone field in the ZMO3 runs *is a function of 3D advection*, which means that the zonal mean ozone field that is passed to the radiation code in the ZMO3 runs does (implicitly) include effects of three-dimensional ozone. This effect is "pathway two" in the nomenclature of the Albers/Nathan paper. Thus in order for the ZMO3 runs to suppress the "pathway two" effects, then the impact of 3D advection processes *cannot* be allowed to alter the ZMO3 ozone field. To be clear, I am fine with whatever setup the authors used, I just think it is very important for the authors to detail *exactly* what types of physics are included in their two run types.

---

## Author Response (AR2)

**Radiative effects of ozone waves on the Northern Hemisphere polar vortex and its modulation by the QBO**

Vered Silverman[1], Nili Harnik[1], Katja Matthes[2], Sandro W. Lubis[3], and Sebastian Wahl[2]

[1]Department of Geophysical, Atmospheric and Planetary Sciences, Tel Aviv University, Tel Aviv, Israel
[2]GEOMAR Helmholtz Centre for Ocean Research Kiel, Kiel, Germany
[3]Department of the Geophysical Sciences, The University of Chicago, USA

*Correspondence to:* Vered Silverman (veredsil@post.tau.ac.il)

**Dear William Ward,**

- – We thank reviewer #1 for pointing out what needs further clarification. As we state in the introduction (page 2 lines 9-15), we concentrate in our paper only on the radiative pathway and not on the advective ozone wave pathway. The reviewer is correct in pointing out that if we use the full 3d ozone field in the ozone advection scheme we are including the effect of ozone waves on the zonal mean ozone field and thus on our dynamics. Since we use the full ozone field in the ozone dynamics and specifically in ozone advection, and we only zonalize ozone when inputing it into the radiation scheme, we are indeed only isolating the influence of the radiative pathway, and not isolating the total effect of ozone waves. We further clarified this point in page 4, lines 25-30, and took care to go over the manuscript and add "radiative effect of ozone waves" when relevant.

- – We thank reviewer #2 for recommending our manuscript for publication.

[revised manuscript text omitted]